# Multiple sources of slow activity fluctuations in a bacterial chemosensory network

**Remy Colin[1,2][†]\*, Christelle Rosazza[1,2][†], Ady Vaknin[3], Victor Sourjik[1,2]\***

[1]Max Planck Institute for Terrestrial Microbiology, Marburg, Germany; [2]LOEWE Center for Synthetic Microbiology (SYNMIKRO), Marburg, Germany; [3]The Racah Institute of Physics, The Hebrew University, Jerusalem, Israel

**Abstract** Cellular networks are intrinsically subject to stochastic fluctuations, but analysis of the resulting noise remained largely limited to gene expression. The pathway controlling chemotaxis of *Escherichia coli* provides one example where posttranslational signaling noise has been deduced from cellular behavior. This noise was proposed to result from stochasticity in chemoreceptor methylation, and it is believed to enhance environment exploration by bacteria. Here we combined single-cell FRET measurements with analysis based on the fluctuation-dissipation theorem (FDT) to characterize origins of activity fluctuations within the chemotaxis pathway. We observed surprisingly large methylation-independent thermal fluctuations of receptor activity, which contribute to noise comparably to the energy-consuming methylation dynamics. Interactions between clustered receptors involved in amplification of chemotactic signals are also necessary to produce the observed large activity fluctuations. Our work thus shows that the high response sensitivity of this cellular pathway also increases its susceptibility to noise, from thermal and out-of-equilibrium processes.

DOI: https://doi.org/10.7554/eLife.26796.001

**\*For correspondence:**
remy.colin@synmikro.mpi-marburg.mpg.de (RC);
victor.sourjik@synmikro.mpi-marburg.mpg.de (VS)

[†]These authors contributed equally to this work

**Competing interests:** The authors declare that no competing interests exist.

## Introduction

It is well established that cellular processes are intrinsically stochastic and therefore prone to fluctuations (*ten Wolde et al., 2016*; *Rao et al., 2002*; *Tsimring, 2014*). The best-characterized examples of cellular noise relate to the variability in expression of genes or proteins, observed either across a population of genetically identical cells or within one cell over time (*Raj and van Oudenaarden, 2008*; *Elowitz et al., 2002*). The molecular origins and physiological effects of such expression noise are comparatively well understood (*Rao et al., 2002*; *Tsimring, 2014*; *Raj and van Oudenaarden, 2008*; *Paulsson, 2004*; *Balázsi et al., 2011*; *Eldar and Elowitz, 2010*). In contrast, noise that arises in cellular networks at the posttranslational level remains much less characterized. Although such noise is expected to be ubiquitous, for example, in signaling networks, it was mostly observed indirectly through its effects on gene expression or cell behavior (*ten Wolde et al., 2016*; *Tsimring, 2014*).

Chemotaxis of *Escherichia coli*, a bacterial model for signal transduction, previously provided one example where signaling noise has been predicted based on analyses of cell motility and flagellar rotation (*Korobkova et al., 2004*; *Emonet and Cluzel, 2008*; *Berg and Brown, 1972*; *Dufour et al., 2016*; *Spudich and Koshland, 1976*; *Park et al., 2010*; *He et al., 2016*). *E. coli* swims by a succession of straight runs during which the bacterium advances, that are interrupted by short reorientations, or tumbles, which results in a random walk. In chemical gradients, this random walk becomes biased by lengthening the runs towards more favorable conditions. The chemotaxis pathway controlling this behavior is composed of two modules, one mediating signal transduction and

another adaptation, that operate on different time scales (*Parkinson et al., 2015*; *Colin and Sourjik, 2017*; *Shimizu et al., 2010*) (*Figure 1—figure supplement 1A*). The signal transduction module includes sensory complexes consisting of the dimers of transmembrane receptors, the kinase CheA and the scaffold protein CheW. Signaling by these complexes can be understood in terms of a two-state model: In the absence of stimulation, receptor dimers are at equilibrium between the active (ON) and inactive (OFF) states, resulting in an intermediate level of autophosphorylation activity of the receptor-associated CheA. Positive chemotactic stimuli (attractants) shift the equilibrium towards the OFF state, thus inhibiting CheA, whereas repellent stimulation has the opposite effect. Downstream signal transduction occurs via phosphorylation of the response regulator CheY that can subsequently bind to the flagellar motors to induce tumbles. CheY is dephosphorylated with the help of the phosphatase CheZ. All reactions within the signal transduction module occur within a few hundred milliseconds (*Sourjik and Berg, 2002a*), ensuring that swimming bacteria can faithfully monitor their current local environment.

The adaptation module operates on a much longer time scale of seconds to minutes. It includes two enzymes, the methyltransferase CheR and the methylesterase CheB, which add or remove respectively methyl groups at four specific glutamyl residues of the chemoreceptors. Since receptor methylation increases the activity of the chemosensory complexes, these changes gradually compensate for the effects of both attractant and repellent stimulation via a negative feedback loop (*Barkai and Leibler, 1997*; *Hansen et al., 2008*; *Tu et al., 2008*). This enables bacteria to robustly maintain an intermediate steady-state activity of CheA, and thus the level of CheY phosphorylation and frequency of cell tumbles, even in the presence of steady background stimulation. Notably, in both major *E. coli* chemoreceptors Tar and Tsr, two of the four methylated residues are initially encoded as glutamines, for example Tar is expressed as Tar$^{QEQE}$. Glutamines are functionally similar to methylated glutamates (*Dunten and Koshland, 1991*; *Sourjik and Berg, 2004*; *Li and Weis, 2000*; *Endres et al., 2008*), and they are subsequently deamidated to glutamates by CheB (*Rice and Dahlquist, 1991*; *Kehry et al., 1983*).

Despite this importance of the adaptation system for robust maintenance of the average signaling output, it was suggested that the relatively small number of methylation enzymes (*Li and Hazelbauer, 2004*) and their slow exchange rates at their receptor substrates (*Li and Hazelbauer, 2005*; *Schulmeister et al., 2008*) lead to fluctuations of the level of phosphorylated CheY (*Korobkova et al., 2004*; *Emonet and Cluzel, 2008*; *Dufour et al., 2016*; *Tu and Grinstein, 2005*; *Pontius et al., 2013*). Further amplified by the cooperative response of the flagellar motor (*Tu and Grinstein, 2005*; *Cluzel et al., 2000*), these fluctuations were proposed to explain the observed large variation in the motor rotation (*Korobkova et al., 2004*; *Emonet and Cluzel, 2008*; *He et al., 2016*) and in the swimming behavior (*Korobkova et al., 2004*; *Berg and Brown, 1972*; *Spudich and Koshland, 1976*; *Taute et al., 2015*) of individual cells over time. Subsequent theoretical analyses suggested that such behavioral fluctuations might provide physiological benefit, by enhancing environmental exploration (*Emonet and Cluzel, 2008*; *Viswanathan et al., 1999*; *Matthäus et al., 2009*; *Bénichou et al., 2011*; *Matthäus et al., 2011*; *Flores et al., 2012*).

Another distinctive feature of the bacterial chemotaxis pathway is the clustering of chemoreceptors in large signaling arrays, formed through a complex network of interactions between trimers of receptor dimers, CheA and CheW (*Parkinson et al., 2015*). Although signaling arrays are stable on the time scale of signal transduction (*Schulmeister et al., 2008*; *Gegner et al., 1992*), they appear to locally reorganize within minutes (*Frank and Vaknin, 2013*). Within arrays, the activity states of neighboring receptors are coupled, resulting in amplification and integration of chemotactic signals (*Sourjik and Berg, 2004*; *Li and Weis, 2000*; *Tu, 2013*; *Piñas et al., 2016*; *Duke and Bray, 1999*; *Mello and Tu, 2003*; *Monod et al., 1965*; *Keymer et al., 2006*). These allosteric receptor interactions have been previously described using either the Monod-Wyman-Changeux (MWC) model (*Monod et al., 1965*) which assumes that receptors operate in units (signaling teams) of 10–20 dimers where activities of individual receptors are tightly coupled (*Sourjik and Berg, 2004*; *Mello and Tu, 2003*; *Monod et al., 1965*; *Keymer et al., 2006*; *Mello and Tu, 2005*) or using an Ising model of a receptor lattice with intermediate coupling (*Duke and Bray, 1999*; *Mello and Tu, 2003*). In both models, the sensitivity of signaling arrays is highest at intermediate levels of receptor activity where receptors can easily switch between ON and OFF states, with optimal intermediate activity being maintained by the adaptation system (*Tu, 2013*; *Piñas et al., 2016*). Another connection between the adaptation system and receptor clustering is through adaptation assistance

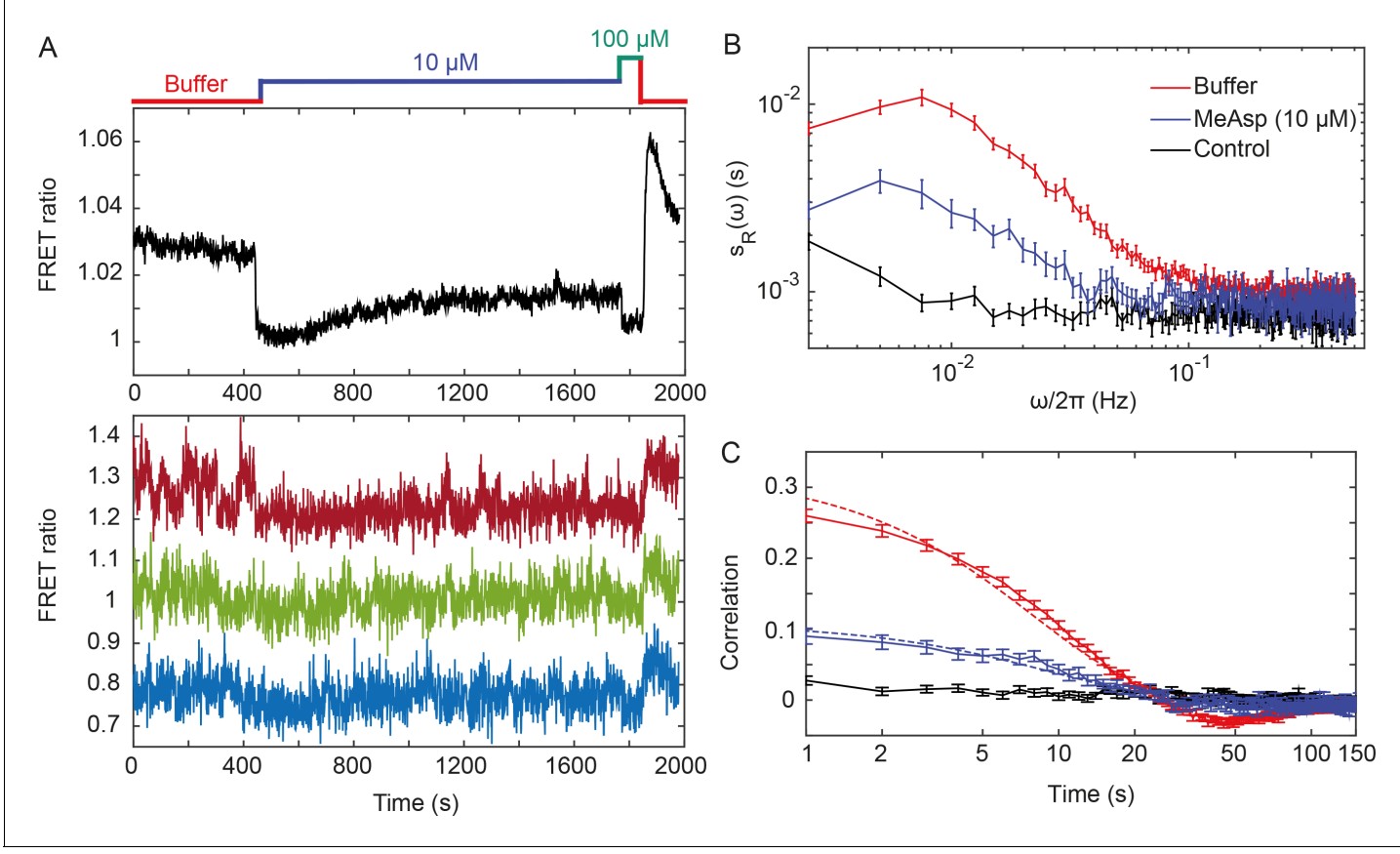

**Figure 1.** Fluctuations of the chemotaxis pathway activity in individual CheR[+]CheB[+] cells. (**A**) Time course of the FRET measurements for the CheR[+] CheB[+] strain expressing the FRET pair CheY-YFP and CheZ-CFP and Tar as the sole receptor (see Materials and methods for details of expression), for cell population (upper panel) and for representative single cells (lower panel). Cells immobilized in a flow chamber under steady flow (see Materials and methods and *Figure 1—figure supplement 1B*) were initially adapted in buffer (red) and subsequently stimulated by addition and subsequent removal of indicated concentrations of a non-metabolizable chemoattractant MeAsp (blue and green). The measurement traces for single cells have been shifted along the *y*-axis to facilitate visualization. (**B**) Power spectral density (PSD) of the FRET ratio for single cells adapted in buffer (red curve) or in 10 μM MeAsp (blue curve), as well as for the control receptorless strain in buffer (black curve). (**C**) The corresponding time autocorrelation functions of the single-cell FRET ratio. Dashed lines show fits by exponential decay (see Materials and methods). The error bars represent standard errors of the mean (SEM), and the sample sizes are 265 (buffer), 69 (10 μM) and 103 (receptorless control) single cells coming from at least three independent experiments in each case.

DOI: https://doi.org/10.7554/eLife.26796.002

The following figure supplements are available for figure 1:

**Figure supplement 1.** Schematic representation of the FRET experiment.

DOI: https://doi.org/10.7554/eLife.26796.003

**Figure supplement 2.** Additional FRET measurement for CheR[+]CheB[+] cells.

DOI: https://doi.org/10.7554/eLife.26796.004

**Figure supplement 3.** Negative controls.

DOI: https://doi.org/10.7554/eLife.26796.005

**Figure supplement 4.** Additional analyses for CheR[+]CheB[+] cells.

DOI: https://doi.org/10.7554/eLife.26796.006

**Figure supplement 5.** Correction of the PSDs for CheR[+]CheB[+] cells for measurement noise.

DOI: https://doi.org/10.7554/eLife.26796.007

neighborhoods, where adaptation enzymes that are transiently tethered to one receptor molecule can methylate (or demethylate) multiple neighboring receptors (*Li and Hazelbauer, 2005*).

In this work we directly quantify signaling noise in *E. coli* chemotaxis, using Förster (fluorescence) resonance energy transfer (FRET) to monitor pathway activity in single cells and with high time resolution. We show that the pathway activity fluctuations arise from interplay of multiple factors,

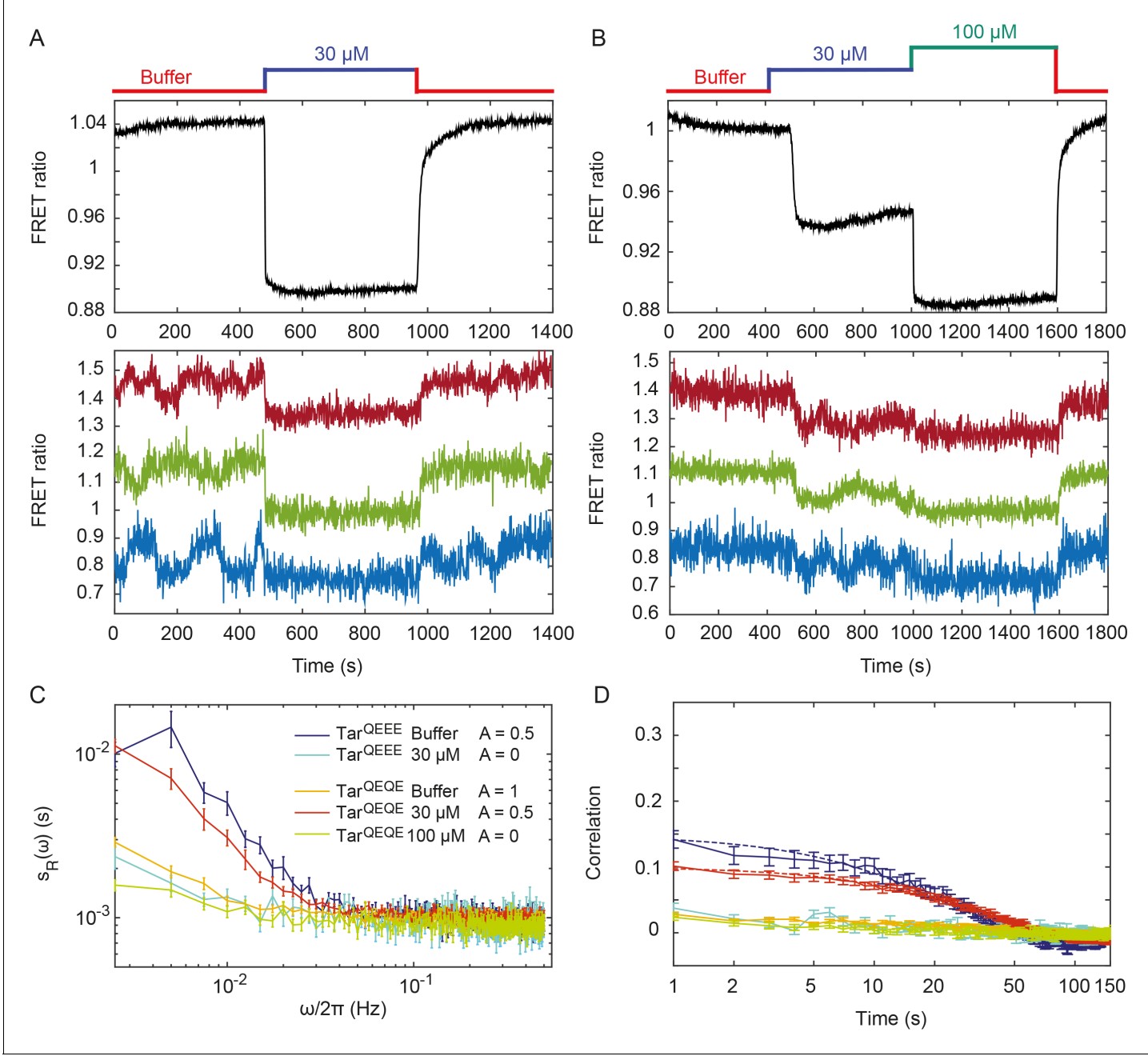

**Figure 2.** Pathway activity fluctuations in Δ*cheR*Δ*cheB* cells. (**A**) Time course of population-averaged (black; upper panel) and typical single-cell (colors; lower panel) measurements of the FRET ratio for Δ*cheR*Δ*cheB* strain expressing Tar^QEEE as the sole receptor. Measurements were performed as in *Figure 1*. Cells were first equilibrated in buffer (red) and subsequently stimulated by addition (blue) and subsequent removal of 30 μM MeAsp, saturating stimulus for this receptor. (**B**) Same as (**A**) but for Δ*cheR*Δ*cheB* strain expressing Tar^QEQE as the sole receptor and upon stimulation with 30 μM (blue) and then 100 μM (green) MeAsp. Note that for this receptor, 30 μM MeAsp is the sub-saturating stimulus whereas 100 μM MeAsp is the saturating stimulus. The measurement traces for single cells in (**A**) and (**B**) have been shifted along the *y*-axis to facilitate visualization. (**C**) PSD of the single-cell FRET ratio for Tar^QEEE in buffer (blue) or in 30 μM MeAsp (cyan), Tar^QEQE in buffer (orange), in 30 μM MeAsp (red) or in 100 μM MeAsp (green). (**D**) Corresponding time autocorrelation functions of the single-cell FRET ratio for indicated strains/conditions. Dashed lines show fits by single exponential decay. Error bars represent standard errors of the mean (SEM), and the sample sizes are 153 (Tar^QEEE, buffer), 65 (Tar^QEEE, 30 μM), 471 (Tar^QEQE, buffer), 404 (Tar^QEQE, 30 μM) and 136 (Tar^QEQE, 100 μM) single cells coming from at least three independent experiments in each case.

DOI: https://doi.org/10.7554/eLife.26796.008

The following figure supplements are available for figure 2:

**Figure supplement 1.** Correction of the PSDs for Δ*cheR*Δ*cheB* cells for measurement noise.

*Figure 2 continued on next page*

*Figure 2 continued*

DOI: https://doi.org/10.7554/eLife.26796.009

**Figure supplement 2.** Comparison of $\Delta cheR\Delta cheB$ and $CheR^+CheB^+$ power spectra.

DOI: https://doi.org/10.7554/eLife.26796.010

including not only the stochasticity of the methylation system but also cooperative interactions and slow rearrangements of receptors within clusters. Finally, using analysis based on the fluctuation-dissipation theorem (FDT) we could distinguish between equilibrium and out-of-equilibrium fluctuations within the chemotaxis network and elucidate respective contributions of receptor clusters and methylation to the overall noise.

## Results

### Fluctuations of chemotaxis pathway activity in single cells

To perform time-resolved measurements of the chemotaxis pathway activity in individual *E. coli* cells, we adapted the microscopy-based ratiometric FRET assay (*Sourjik et al., 2007*) that relies on the phosphorylation-dependent interaction between CheY, fused to yellow fluorescent protein (CheY-YFP), and its phosphatase CheZ, fused to cyan fluorescent protein (CheZ-CFP) (*Figure 1—figure supplement 1A*). The amount of this complex, and thus the level of FRET, provides a direct intracellular readout of CheA activity (*Endres et al., 2008*; *Sourjik et al., 2007*; *Sourjik and Berg, 2002b*; *Oleksiuk et al., 2011*). In previous studies where this assay was applied to investigate chemotactic signaling in *E. coli* populations (*Shimizu et al., 2010*; *Sourjik and Berg, 2004*; *Endres et al., 2008*; *Frank and Vaknin, 2013*; *Piñas et al., 2016*; *Sourjik et al., 2007*; *Sourjik and Berg, 2002b*; *Oleksiuk et al., 2011*; *Neumann et al., 2014a*; *Krembel et al., 2015a*; *Meir et al., 2010*; *Frank et al., 2016*; *Krembel et al., 2015b*; *Neumann et al., 2014b*; *Neumann et al., 2012*; *Clausznitzer et al., 2010*), bacteria expressing the FRET pair were immobilized in a flow chamber and fluorescent signals were collected using photon counters from an area containing several hundred cells (*Sourjik et al., 2007*). Here, we used a similar setup but instead imaged fluorescence of the FRET pair with an electron multiplication charge-coupled device (EM-CCD) camera (see Materials and methods and *Figure 1—figure supplement 1B,C*).

As done previously (*Sourjik and Berg, 2004*; *Endres et al., 2008*; *Oleksiuk et al., 2011*; *Meir et al., 2010*), we analyzed *E. coli* cells that express the CheY-YFP/CheZ-CFP FRET pair instead of the native CheY and CheZ and have Tar as the only chemoreceptor (see Materials and methods). The level of Tar expression in these cells and under our conditions is ~$10^4$ dimers per cell (*Sourjik and Berg, 2004*; *Endres et al., 2008*), comparable to the total level of endogenous chemoreceptors (*Li and Hazelbauer, 2004*). When integrated over the population, the chemotactic response of these cells measured using EM-CCD (*Figure 1A* and *Figure 1—figure supplement 2*, upper panel) was very similar to the one observed previously using area detectors (*Sourjik and Berg, 2002b*; *Meir et al., 2010*). When bacteria in the flow chamber were stimulated with the Tar-specific chemoattractant α-methyl-DL-aspartate (MeAsp), the ratio of the YFP to CFP fluorescence (FRET ratio, $R(t) = YFP(t)/CFP(t)$) first rapidly decreased. This is consistent with the fast attractant-mediated inhibition of the kinase activity, which results in decreased formation of the FRET complex, and therefore reduced energy transfer from the donor (CFP) to the acceptor (YFP) fluorophore. As 10 μM MeAsp is known to fully inhibit the kinase activity in this strain (*Sourjik and Berg, 2004*; *Endres et al., 2008*), the value of the FRET ratio immediately after stimulation reflects the zero activity baseline. Subsequently, the pathway adapted to the new background level of attractant via the CheR-dependent increase in receptor methylation. But as previously reported adaptation of Tar-only cells to high levels of MeAsp was only partial (*Neumann et al., 2014a*; *Krembel et al., 2015a*; *Meir et al., 2010*), meaning that the adapted pathway activity remained lower than in buffer. Subsequent removal of attractant resulted in a transient increase in kinase activity, followed by the CheB-mediated adaptation through the demethylation of receptors.

Although the FRET ratio measured for individual cells during the same experiment was expectedly noisier than the population-averaged data, both the initial response and subsequent adaptation were clearly distinguishable (*Figure 1A* and *Figure 1—figure supplement 2*, lower panel). In

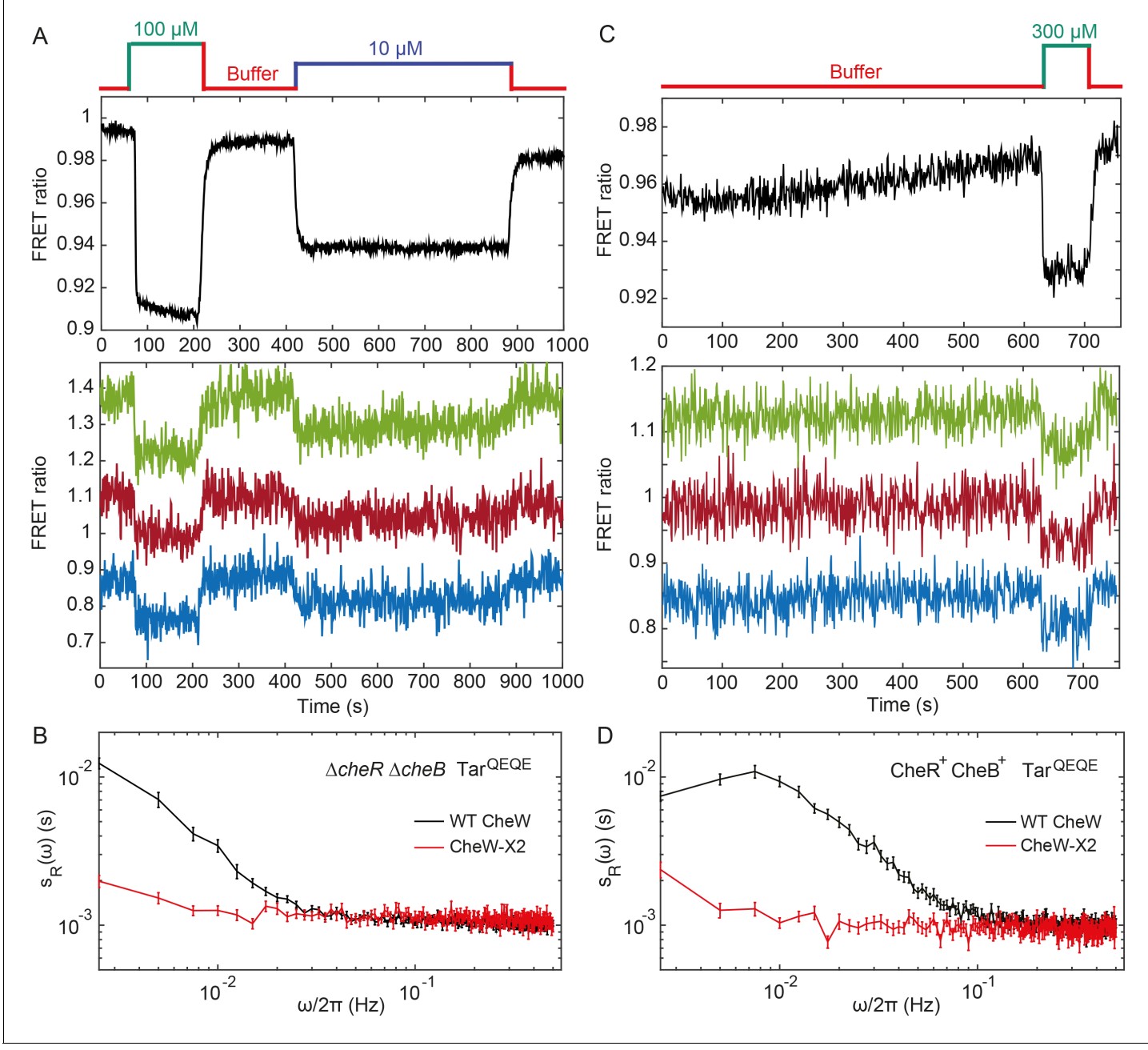

**Figure 3.** Fluctuation analysis in CheW-X2 cells. (A) Population-averaged (upper panel) and typical single-cell (lower panel) measurements of the FRET ratio for $\Delta cheR\Delta cheB$ strain carrying CheW-X2 and Tar$^{QEQE}$ as the sole receptor. Cells, which have a high activity in buffer, were first exposed to 100 µM MeAsp (saturating stimulus), and then to 10 µM MeAsp (sub-saturating stimulus), as indicated. The single-cell measurement traces have been shifted along the *y*-axis to facilitate visualization. (B) Power spectral density of the FRET ratio fluctuations in CheW-X2 $\Delta cheR\Delta cheB$ Tar$^{QEQE}$ cells at intermediate activity (i.e., with 10 µM MeAsp) (red) compared to the equivalent strain carrying native (wild-type; WT) CheW and at 30 µM MeAsp (black – same data as *Figure 2C*). Error bars represent SEM, with sample sizes 404 (WT CheW; black) and 208 (CheW-X2; red) cells. (C) Same as (A) but for CheR$^+$ CheB$^+$ strain. The activity in buffer is at intermediate level (*Figure 3—figure supplement 2*), with 300 µM MeAsp completely inhibiting the kinase activity. (D) Power spectral density of the FRET ratio fluctuations in CheR$^+$ CheB$^+$ CheW-X2 strain in buffer (red) compared to the native WT CheW (black – same data as *Figure 1C*). Error bars represent SEM, with sample sizes 265 (WT CheW; black) and 191 (CheW-X2; red) cells.

DOI: https://doi.org/10.7554/eLife.26796.011

The following figure supplements are available for figure 3:

**Figure supplement 1.** Dose response to MeAsp of $\Delta cheR\Delta cheB$ CheW-X2 cells expressing Tar$^{QEQE}$.

DOI: https://doi.org/10.7554/eLife.26796.012

**Figure supplement 2.** Response of CheR$^+$CheB$^+$ strain expressing CheW-X2 and Tar$^{QEQE}$ to attractant MeAsp and repellent Ni$^{2+}$.

*Figure 3 continued on next page*

*Figure 3 continued*

DOI: https://doi.org/10.7554/eLife.26796.013

**Figure supplement 3.** Response function for ΔcheRΔcheB CheW-X2 cells expressing Tar$^{QEQE}$ as sole receptor.

DOI: https://doi.org/10.7554/eLife.26796.014

contrast to the population measurement, however, a majority of individual cells also exhibited large fluctuations in the FRET ratio on the time scale of 10–100 s. For cells adapted in buffer, the amplitude of these fluctuations could be as large as the response to strong attractant stimulus. Confirming that this low-frequency noise reflects fluctuations of the pathway activity, it was not observed when imaging either fluorescent beads or the same FRET pair in receptorless cells that do not activate CheA (*Figure 1—figure supplement 3A,B*). Furthermore, inhibition of the pathway activity by saturating stimulation with 10 μM or 25 μM MeAsp also transiently suppressed long-term fluctuations, which subsequently (partly) reappeared upon (partial) recovery of the pathway activity due to adaptation (*Figure 1A* and *Figure 1—figure supplement 2*). In contrast, the higher-frequency noise in the FRET ratio could be observed in all strains and conditions, including receptorless cells, indicating that it represents the noise of the measurement. High-frequency noise was also observed in the control measurements using fluorescent beads, although its magnitude was lower, consistent with higher brightness of beads compared to the YFP/CFP expressing cells.

To analyze these activity fluctuations in greater detail, we computed the power spectral density (PSD) of the single-cell FRET ratio, $s_R(\omega)$ (see Materials and methods). The PSD extracts the average spectral content of the temporal variations of the single-cell FRET ratio, that is determines the frequencies at which this ratio fluctuates, with $s_R(\omega)$ representing the magnitude of fluctuations at a given frequency $\omega$. We observed that at high frequency ($\omega > 0.1$ Hz) the PSD kept a constant frequency-independent low value that was similar in all strains (*Figure 1B*). We thus conclude that the noise in the FRET ratio in this frequency range is dominated by the shot noise of the measurement. At lower frequency, however, the PSD measured for the Tar-expressing cells adapted in buffer increased dramatically (roughly as $1/\omega$), reaching a low frequency plateau at $\omega/2\pi \simeq 0.015$ Hz. A similar result was obtained for cells expressing Tar in the unmodified (Tar$^{EEEE}$) state, where all glutamates are directly available for methylation by CheR (*Figure 1—figure supplement 4A*). The increase of the PSD at low frequency was also observed for cells adapted to either 10 or 25 μM MeAsp (*Figure 1B* and *Figure 1—figure supplement 4A*), although the amplitude of this increase was smaller than for the buffer-adapted cells, apparently consistent with their lower pathway activity (*Figure 1A* and *Figure 1—figure supplement 2*). The receptorless strain showed nearly constant noise level over the entire frequency range, as expected for white shot noise, although the PSD increased weakly at the lowest frequency. As such increase was not observed for the control using fluorescent beads (*Figure 1—figure supplement 3A*), it might be due to the slow drift of the FRET ratio arising as a consequence of the slightly different bleaching rates of CFP and YFP, but possibly also to slow changes in cell physiology. In any case, the contribution of this low-frequency component to the overall PSD of the Tar-expressing cells is only marginal (note the log scale in *Figure 1B*), and subtracting it did not markedly change our results (*Figure 1—figure supplement 5*).

The PSD was further used to calculate the average time autocorrelation function of the single-cell FRET ratio, which reflects the characteristic time scale of activity fluctuations (see Materials and methods). For cells adapted in buffer, the autocorrelation time constant was 9.5 ± 0.5 s, as determined by an exponential fit to the autocorrelation function (*Figure 1D*). This value is similar to the characteristic time of the pathway activity fluctuation previously deduced from behavioral studies (*Korobkova et al., 2004*; *Park et al., 2010*). The same characteristic time was observed in MeAsp-adapted cells, although the amplitude of the correlation was considerably smaller in this case (*Figure 1D* and *Figure 1—figure supplement 4B*). Interestingly, at longer times the autocorrelation function becomes weakly negative, indicating an overshoot that is likely caused by the negative feedback in the adaptation system (*Berg and Tedesco, 1975*). As expected, no autocorrelation was observed for the receptorless cells.

Finally, the variance of activity was evaluated from the PSD using Parseval's formula (*Gasquet and Witomski, 1999*). After subtracting the variance measured for the receptorless strain, which reflects the contribution of the shot noise, the specific variance of the FRET ratio for cells

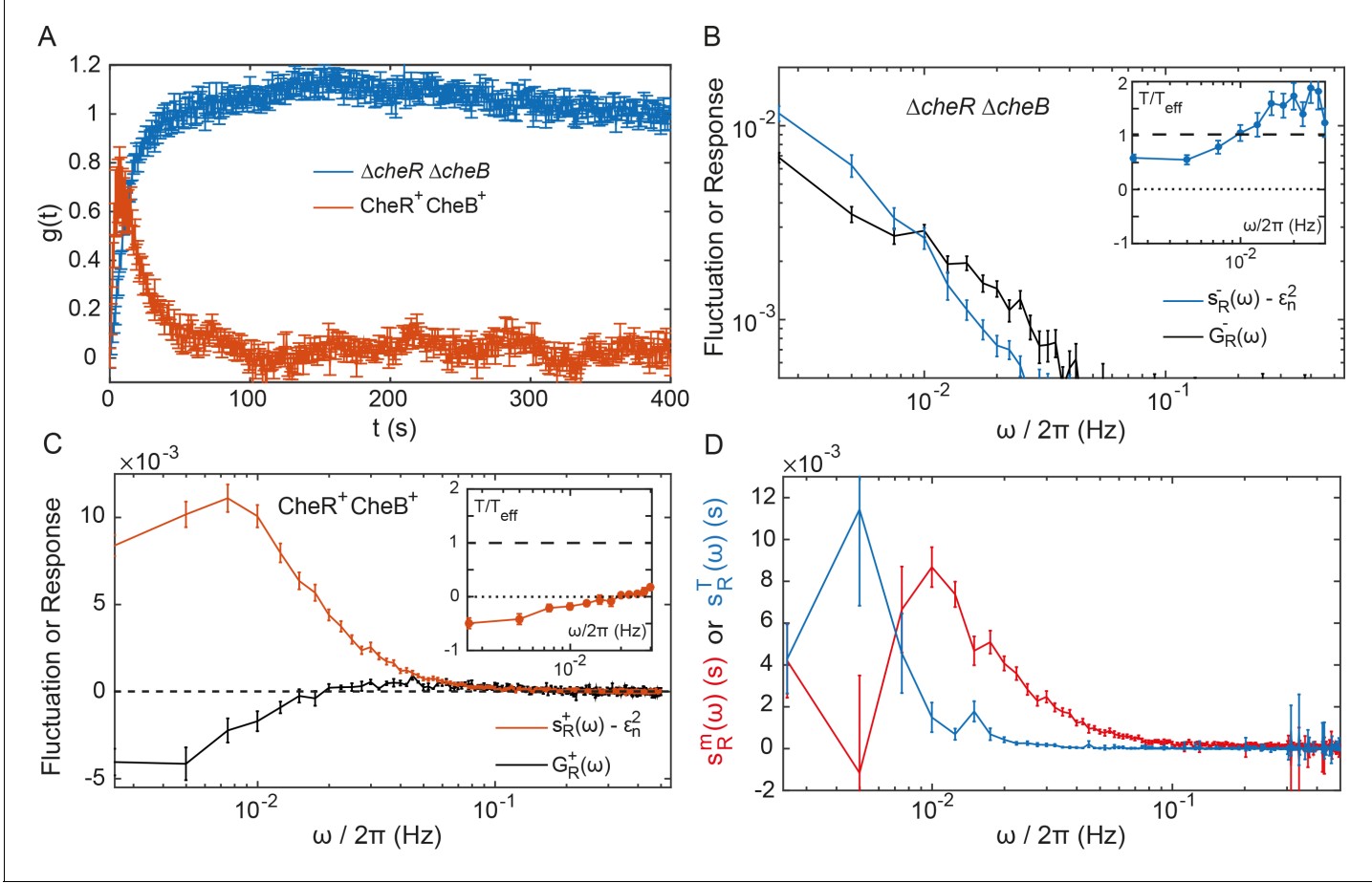

**Figure 4.** Fluctuation-dissipation analysis of the pathway activity. (A) Step response function $g(t)$ both in presence (red) and in absence (blue) of the adaptation enzymes, evaluated in cells expressing Tar$^{QEQE}$ that respond to a step change from buffer to 0.3 μM MeAsp (CheR$^+$ CheB$^+$) or 30 μM MeAsp ($\Delta cheR\Delta cheB$). The step response function was calculated from the measurements shown in *Figure 4—figure supplement 5* and in *Figure 2B* as described in text and in Appendix 1, 'Phenomenological step response function'. (B,C) The PSD of the FRET ratio fluctuations $s_R(\omega)$ at $\langle A \rangle = 0.5$ (blue in <u>B</u> and red in <u>C</u>), and the corresponding dissipation $G_R(\omega)$ (black) calculated using *Equation (2)*, for $\Delta cheR\Delta cheB$ (B) and CheR$^+$CheB$^+$ (C) cells. The measurement shot noise $\epsilon_n^2$, determined as the PSD of the receptorless cells (*Figure 1B*), was subtracted from $s_R(\omega)$. *Insets* show the ratio between the physical and effective temperatures, calculated using *Equation (1)*. Dashed and dotted lines in <u>B</u> and <u>C</u> indicate $T/T_{\text{eff}}(\omega) = 1$ and $T/T_{\text{eff}}(\omega) = 0$, respectively. (D) Contribution of thermal noise (blue) and the adaptation enzyme dynamics (red) to the PSD in CheR$^+$ CheB$^+$ cells, calculated from *Equation (3)* as explained in Appendix 1, 'Separating the contribution of methylation enzymes dynamics to the PSD in CheR$^+$ CheB+ cells'. In all panels, error bars represent SEM, with sample sizes for the power spectra calculations being 540 ($\Delta cheR\Delta cheB$) and 468 (CheR$^+$ CheB$^+$; aggregating data from cells expressing Tar$^{QEQE}$ and Tar$^{EEEE}$ as sole receptor) single cells from at least five biological replicates.

DOI: https://doi.org/10.7554/eLife.26796.015

The following figure supplements are available for figure 4:

**Figure supplement 1.** Validation of the linear response regime in $\Delta cheR\Delta cheB$ cells.
DOI: https://doi.org/10.7554/eLife.26796.016

**Figure supplement 2.** Power spectral density computed on subsets of the cell populations sorted according to their activity.
DOI: https://doi.org/10.7554/eLife.26796.017

**Figure supplement 3.** Effect of the receptor expression level on the noise in $\Delta cheR\Delta cheB$ cells.
DOI: https://doi.org/10.7554/eLife.26796.018

**Figure supplement 4.** Power spectra of thermal fluctuations in a simulated model of the sensory cluster.
DOI: https://doi.org/10.7554/eLife.26796.019

**Figure supplement 5.** Example of FRET measurement used for the evaluation of $g^+(t)$.
DOI: https://doi.org/10.7554/eLife.26796.020

**Figure supplement 6.** Evaluation of the adaptation time.
DOI: https://doi.org/10.7554/eLife.26796.021

**Figure supplement 7.** Calculated effective temperatures.

*Figure 4 continued on next page*

*Figure 4 continued*

DOI: https://doi.org/10.7554/eLife.26796.022

**Figure supplement 8.** Inferred spectrum of the binding dynamics of CheR and CheB.

DOI: https://doi.org/10.7554/eLife.26796.023

**Figure supplement 9.** Simulation of the pathway activity fluctuations in adapting cells.

DOI: https://doi.org/10.7554/eLife.26796.024

adapted in buffer was $\langle \Delta R^2 \rangle^+ = 0.0046 \pm 0.0002$ (where '+' refers to the presence of adaptation enzymes, CheR$^+$ CheB$^+$). As shown previously (*Sourjik et al., 2007*), the FRET ratio $R$ is related to the relative pathway activity $\langle A \rangle$ as $R = \lambda \langle A \rangle + \mu$, where $\lambda$ is the conversion factor and $\mu$ is a constant corresponding to the baseline FRET ratio at zero pathway activity (i.e., upon stimulation with saturating attractant concentration; *Figure 1A*). The value $\lambda = 0.10 \pm 0.01$ could be estimated as the mean difference between the measured FRET ratio values corresponding to the fully active (i.e., $\langle A \rangle = 1$) and fully inactive (i.e., $\langle A \rangle = 0$) pathway (see Materials and methods). The calculated variance of the pathway activity was thus $\langle \Delta A^2 \rangle^+ = 0.46 \pm 0.04$, indicating concerted activity fluctuations across much of the signaling array.

## Activity fluctuations in absence of adaptation system

We next monitored the single-cell pathway activity in a strain lacking CheR and CheB, to test whether the observed fluctuations could be solely explained by the action of the adaptation system. Given the observed dependence of the fluctuations on the level of pathway activity, we first analyzed a $\Delta cheR \Delta cheB$ strain that was engineered to express Tar receptor in one-modified state (Tar$^{QEEE}$). This closely mimics the average modification state and intermediate activity of Tar in CheR$^+$ CheB$^+$ cells adapted in buffer (*Endres et al., 2008*; *Sourjik and Berg, 2002b*). Expectedly, $\Delta cheR \Delta cheB$ Tar$^{QEEE}$ cells responded to MeAsp but showed no adaptation comparable to CheR$^+$ CheB$^+$ cells (*Figure 2A*). But despite the lack of the adaptation system, pathway activity in individual $\Delta cheR \Delta cheB$ Tar$^{QEEE}$ cells showed pronounced long-term fluctuations when cells were equilibrated in buffer (*Figure 2A*, lower panel). These methylation-independent long-term fluctuations were suppressed upon saturating pathway inhibition with 30 μM MeAsp, leaving only the shot noise of the measurement.

In contrast to Tar$^{QEEE}$, $\Delta cheR \Delta cheB$ cells expressing the half-modified Tar$^{QEQE}$ as the sole receptor showed no long-term activity fluctuations in buffer (*Figure 2B*). Because Tar$^{QEQE}$ is known to be highly active (*i.e.*, strongly biased towards the ON state) in absence of attractants (*Endres et al., 2008*; *Oleksiuk et al., 2011*) and therefore insensitive to stimulation, we lowered its activity to an intermediate value by stimulating cells with 30 μM MeAsp (*Figure 2B*, upper panel). This partial stimulation indeed restored low-frequency fluctuations in $\Delta cheR \Delta cheB$ Tar$^{QEQE}$ cells (*Figure 2B*, lower panel). Again, these activity fluctuations were completely abolished upon saturating attractant stimulation. Cumulatively, these results clearly demonstrate that, at intermediate level of activity where the receptors are highly sensitive, pathway output fluctuates even in the absence of the methylation system. These fluctuations were clearly identifiable above shot noise in the PSD of the FRET ratio (*Figure 2C*), and they were absent under conditions of very low or very high activity. Notably,

**Table 1.** Parameters of the FDT analysis

| Parameter | Value |
| --- | --- |
| $\lambda$ | $0.10 \pm 0.01$ |
| $N$ | 14 |
| $\langle A \rangle$ | 0.5 |
| $X_{A^\infty}$ | $N \langle A \rangle (1 - \langle A \rangle)$ |
| $N_T$ | $10^4$ |
| $\epsilon_n^2$ | $0.9 \ 10^{-3}$ |

DOI: https://doi.org/10.7554/eLife.26796.025

these methylation-independent fluctuations were slower than those observed in CheR$^+$ CheB$^+$ cells (*Figure 2—figure supplement 2*), with a typical time scale of $34 \pm 4$ s, as determined by fitting the time autocorrelation functions with an exponential decay (*Figure 2D*), although this time might be slightly under-evaluated since it is already comparable to the total duration of acquisition (400 s). Their amplitude, evaluated again using Parseval's formula, was $\langle \Delta R^2 \rangle^- = 0.0025 \pm 0.0001$, corresponding to $\langle \Delta A^2 \rangle^- = 0.25 \pm 0.01$, and thus roughly half of the amplitude of fluctuations observed in CheR$^+$ CheB$^+$ cells.

## Role of receptor clustering in signaling noise

To investigate whether the observed fluctuations depend on clustering of chemotaxis receptors, we utilized a recently described CheW-X2 version of the adaptor protein CheW that disrupts formation of the receptor arrays without abolishing signaling (*Frank et al., 2016*). This CheW mutant carries two amino acid replacements, R117D and F122S, which are believed to break the receptor arrays into smaller complexes consisting of two trimers of receptor dimers coupled to one CheA (*Piñas et al., 2016*; *Frank et al., 2016*). The CheW-X2 is expressed at a level similar to the native CheW (*Piñas et al., 2016*). Consistent with reported functionality of such complexes (*Piñas et al., 2016*; *Frank et al., 2016*; *Li and Hazelbauer, 2014*), a $\Delta cheR \Delta cheB$ strain expressing CheW-X2 and Tar$^{QEQE}$ showed basal activity and response to MeAsp which were similar to the respective strain expressing the native CheW (*Figure 3A* and *Figure 3—figure supplement 1*). Nevertheless, this strain showed no apparent long-term fluctuations in the pathway activity above the shot noise, even when its activity was tuned to an intermediate level by addition of 10 µM MeAsp (*Figure 3A,B*). Similarly, the array disruption allowed signaling but abolished the long-term activity fluctuations in CheR$^+$ CheB$^+$ cells equilibrated in buffer (*Figure 3C,D*). Importantly, buffer-adapted CheR$^+$ CheB$^+$ CheW-X2 cells had intermediate receptor activity and could respond to both attractant (MeAsp) and repellent (Ni$^{2+}$) stimuli, that is, both down- and upregulation of the pathway activity (*Figure 3—figure supplement 2*). This confirms that the observed loss of fluctuations was not caused by locking the receptor in the extreme activity state. In summary, these results demonstrate that the observed long-term fluctuations in activity, seen both with and without the receptor methylation system, require receptor clustering.

## Fluctuation-dissipation relation for receptor clusters

We next used mathematical analysis to better understand the respective contributions of receptor clustering and the methylation enzymes to the observed fluctuations and to determine whether methylation-independent fluctuations are generated by some out-of-equilibrium random process. We considered the fluctuation-dissipation theorem (FDT), which postulates – *for systems at equilibrium* – that thermal fluctuations of a quantity are related, via the temperature, to the response of this quantity to a small externally applied perturbation (*Kubo, 1966*). The FDT framework can be used to determine whether a system is at equilibrium, by comparing fluctuations and responses to small perturbations *via* their ratio, the so-called effective temperature $T_{\text{eff}}(\omega)$ (*Robert et al., 2010*; *Martin et al., 2001*; *Mizuno et al., 2007*; *Cugliandolo, 2011*). In equilibrium systems the FDT is satisfied and $T_{\text{eff}}(\omega)$ equals the physical temperature $T$. In out-of-equilibrium (biological) systems, the deviation of $T_{\text{eff}}(\omega)$ from $T$ provides a first characterization of the underlying out-of-equilibrium noisy process generating the fluctuations, since $T_{\text{eff}}(\omega)$ is linked to the energy scale and frequency content of such process (*Robert et al., 2010*; *Martin et al., 2001*; *Mizuno et al., 2007*; *Cugliandolo, 2011*).

In our case, the magnitude of activity fluctuations could be expressed as the PSD corrected for the measurement shot noise, $s_R(\omega) - \epsilon_n^2$, where $\epsilon_n^2$ was experimentally determined as the PSD of the receptorless cells. We therefore define the effective temperature as:

$$\frac{T}{T_{\text{eff}}(\omega)} = \frac{G_R(\omega)}{s_R(\omega) - \epsilon_n^2}. \tag{1}$$

The dissipation $G_R(\omega)$ could be determined by formulating the fluctuation dissipation relation for the activity of individual receptors within the signaling array, using the Ising-like model (*Duke and Bray, 1999*; *Hansen et al., 2010*; *Shimizu et al., 2003*) to describe cooperative receptor interactions as (see Appendix 1, 'Modeling activity fluctuations in the framework of fluctuation-dissipation relation'):

$$G_R(\omega) = -2\,\lambda^2 \frac{3N^2 \langle A \rangle (1 - \langle A \rangle)}{N_T} \mathrm{Re}(\hat{g}(\omega)). \tag{2}$$

Here $\langle A \rangle$ is the average activity around which fluctuations occur, estimated from experimental data as described above, $N_T$ is the total number of Tar dimers per cell, $N$ is the average number of effectively coupled allosteric signaling units in the cluster, and $\lambda$ is defined as before. Consistent with several recent reports (*Piñas et al., 2016*; *Frank et al., 2016*; *Li and Hazelbauer, 2014*) and with our analysis of the apparent response cooperativity in the CheW-X2 strain (*Figure 3—figure supplement 1* and Appendix 1, 'Definition of the effective temperature'), we assumed that signaling units within the cluster correspond to trimers of receptor dimers. Finally, $\mathrm{Re}(\hat{g}(\omega))$ is the real part of the Fourier transform of the normalized step response function $g(t)$, which could be experimentally determined by measuring the FRET response to sufficiently small (subsaturating) stepwise attractant stimulation as $g(t) = \Delta R(t)/(-\lambda X_A^\infty \epsilon_0)$, where $(-\lambda X_A^\infty \epsilon_0)$ is the normalized stimulation strength (see Appendix 1, 'Phenomenological step response function').

For subsaturating stimulation of the non-adapting $\Delta cheR \Delta cheB$ cells (*Figure 2B*), the normalized step response function $g^-(t)$ exhibited a relatively rapid initial increase and then slowly approached its final value, possibly with a slight transient overshoot (*Figure 4A*). Nearly identical response dynamics was observed for weaker stimulations (*Figure 4—figure supplement 1*), validating the small perturbation assumption of the FDT for this response function measurement. This slow response dynamics is consistent with a previous report that attributed it to gradual stimulation-dependent changes in packing of receptors within clusters (*Frank and Vaknin, 2013*). Consistent with this interpretation, the CheW-X2 $\Delta cheR \Delta cheB$ strain with disrupted receptor clustering showed neither comparable latency nor overshoot in its response (*Figure 3—figure supplement 3*).

As the pathway activity in the CheW-X2 $\Delta cheR \Delta cheB$ strain also showed no long-term fluctuations (*Figure 3B,C*), we hypothesized that these fluctuations might be indeed caused by the slow response dynamics stimulated by some random process. We thus calculated the corresponding dissipation using *Equation (2)*, considering that under our conditions $N_T \sim 10^4$ (*Endres et al., 2008*) and $N \sim 14$ (*Endres et al., 2008*; *Neumann et al., 2014a*; *Neumann et al., 2014b*; *Clausznitzer et al., 2010*) (see *Table 1* for all parameter values). At low frequencies, the dissipation $G_R(\omega)$ was approximately equal to the shot-noise corrected $s_R(\omega)$ at $\langle A \rangle \simeq 0.5$ (*Figure 4B*), as predicted by *Equation (1)* for equilibrium systems where $T_{\mathrm{eff}}(\omega)$ equals $T$. Consistently, the corresponding ratio $T/T_{\mathrm{eff}}(\omega)$ was nearly independent of $\omega$ and close to unity in the range of frequencies for which $s_R(\omega)$ is above the measurement noise (*Figure 4B Inset*). This suggested that in absence of adaptation enzymes the system is close to equilibrium and thermal fluctuations are the major source of noise. Although the deviation of $T/T_{\mathrm{eff}}(\omega)$ from unity might indicate second-order contributions of out-of-equilibrium processes, it is comparable to what was observed for other equilibrium systems with measurement methods of similar precision (*Martin et al., 2001*; *Wang et al., 2006*; *Abou and Gallet, 2004*). Thus, an equilibrium model can fairly accurately describe the details of observed long-time activity fluctuations in $\Delta cheR \Delta cheB$ cells. This agreement suggests that the receptor cluster in these cells largely acts as a passive system, where thermal fluctuations stimulate the long-term response dynamics, possibly due to slow changes in receptor packing within clusters, to generate activity fluctuations.

Furthermore, the PSD of $\Delta cheR \Delta cheB$ cells followed the scaling $\langle A \rangle (1 - \langle A \rangle)$, which is expected from the underlying receptor activity being a two-state variable, as evident for subpopulations of cells sorted according to their activity (*Figure 4—figure supplement 2*), with which our FDT analysis is consistent (*Equation 2*). Fluctuations were apparently unaffected by the expression level of Tar, in the tested range of induction (*Figure 4—figure supplement 3*). In the FDT framework, this implies that $N^2/N_T$ must be constant for varying receptor expression, and previous measurements indeed suggest that the cooperativity rises with the expression level of Tar$^{\mathrm{QEQE}}$ in a way that $N^2/N_T$ remains unchanged (*Endres et al., 2008*).

To evaluate the respective effects of signal amplification and the slow dynamics of the cluster activity response, we performed stochastic simulations of a simple model of sensory complexes without adaptation and under thermal noise (see Appendix 1, 'Simulation of a simplified model for the array of receptors'). In this model, receptors are clustered in signaling teams that respond to allosterically amplified free energy changes on an effective time scale averaging the fast switching dynamics and the slow dynamics of the receptor cluster, which accounts qualitatively for the pathway

behavior. Expectedly, larger amplification led to larger fluctuations, and the time scale of the fluctuations followed the imposed response time scale of the cluster. Less trivially, slower response also led to higher maximal amplitude of the fluctuations (*Figure 4—figure supplement 4*).

## Out-of-equilibrium dynamics in presence of adaptation system

The normalized step response function of CheR$^+$ CheB$^+$ cells, $g^+(t)$ (*Figure 4A*), was determined using weak stimulation by 0.3 µM MeAsp, with the activity change $\Delta A / \langle A \rangle = 0.25$ (*Figure 4—figure supplement 5* and Appendix 1, 'Phenomenological step response function'). Describing adaptation according to the classical two-state models of receptors (*Barkai and Leibler, 1997*; *Clausznitzer et al., 2010*; *Mello and Tu, 2007*), the responses of $\Delta cheR \Delta cheB$ and CheR$^+$ CheB$^+$ cells could be linked via the rate of adaptation $\omega_{RB}$, which yielded $\omega_{RB} = 0.06 \pm 0.01$ Hz (Appendix 1, 'Link between the response functions in $\Delta cheR \Delta cheB$ and CheR$^+$ CheB$^+$ cases' and *Figure 4—figure supplement 6*), consistent with previous estimates (*Park et al., 2010*).

The corresponding dissipation $G_R^+(\omega)$, calculated as above according to *Equation (2)*, differed strongly from the PSD of the activity fluctuations (*Figure 4C*), confirming that the system operates out of equilibrium. The corresponding $T/T_{\mathrm{eff}}(\omega) << 1$ (*Figure 4D Inset*) is consistent with strong out-of-equilibrium drive. It decreased at low frequencies, crossing zero at $\omega/2\pi \simeq 0.015$ Hz where $T_{\mathrm{eff}}(\omega)$ diverges (*Figure 4—figure supplement 7*) and dissipation becomes negative. Such crossing indicates a transition to the range of frequencies where the active process dominates (*Martin et al., 2001*; *Sartori and Tu, 2015*), with the frequency of divergence of $T_{\mathrm{eff}}(\omega)$ representing interplay between the time scales of the passive receptor response and adaptation (Appendix 1, 'Frequency of effective temperature divergence').

To further separate specific contributions of the methylation system and thermally activated receptor cluster rearrangements to the power spectrum of activity fluctuations in CheR$^+$ CheB$^+$ cells, we followed previous modeling approaches (*Clausznitzer and Endres, 2011*; *Sartori and Tu, 2011*; *Aquino et al., 2011*) (Appendix 1, 'Separating the contribution of methylation enzymes dynamics to the PSD in CheR$^+$ CheB$^+$ cells'). Assuming that thermal noise behaves the same in presence and in absence of the methylation system, $s_R^+(\omega)$ can be decomposed into a ''thermal'' contribution $s_R^T(\omega)$ and a contribution of the methylation noise $s_R^m(\omega)$:

$$s_R^+(\omega) = s_R^m(\omega) + s_R^T(\omega) = s_R^m(\omega) + \left| \frac{g^+(\omega)}{g^-(\omega)} \right|^2 s_R^-(\omega). \tag{3}$$

Although relatively noisy, particularly at low frequencies, $s_R^m(\omega)$ inferred from *equation (3)* peaked around $\omega_{peak}/2\pi = 0.01$ Hz (*Figure 4D*), which equals the independently determined adaptation rate (see above), $\omega_{peak} \simeq \omega_{RB} = 0.06$ Hz. The contribution of the thermal noise $s_R^T(\omega)$ had a similar magnitude but dominated at lower frequencies. The power spectrum of the CheR and CheB binding events was inferred from $s_R^m(\omega)$ using the previous model and previous conclusion that the methylation-dependent activity fluctuations mainly arise from the intermittent binding of the small number of CheR and CheB molecules to the receptors (*Pontius et al., 2013*). This spectrum was consistent with the common assumption that CheR (CheB) loads and acts only on the inactive (active) receptor (Appendix 1, 'Separating the contribution of methylation enzymes dynamics to the PSD in CheR$^+$ CheB$^+$ cells' and *Figure 4—figure supplement 8*).

We further extended our simulation model of the receptor array composed of independent signaling teams, to test whether we can reproduce the observed power spectrum in presence of adaptation enzymes. Consistent with the large excess of receptors compared to the methylation enzymes (*Li and Hazelbauer, 2004*), in these simulations only one CheR (or CheB) molecule can bind to the inactive (respectively active) receptor team, methylate (respectively demethylate) the receptors, and unbind once the team has turned active (respectively inactive) (Appendix 1, 'Simulation of a simplified model for the array of receptors'). The simulations agreed qualitatively well with the experiments, including the power spectra of CheR/CheB binding and effective temperature (*Figure 4—figure supplement 9*), although absolute amplitudes of the fluctuations were clearly underestimated by the model, as already observed in a previous theoretical work (*Sartori and Tu, 2015*). The simulation also reproduced the loss of slow fluctuations upon disruption of clusters in CheR$^+$ CheB$^+$ cells, which arises from the dependence of $s_R(\omega)$ on the size $N$ of signaling teams. In contrast, simulating

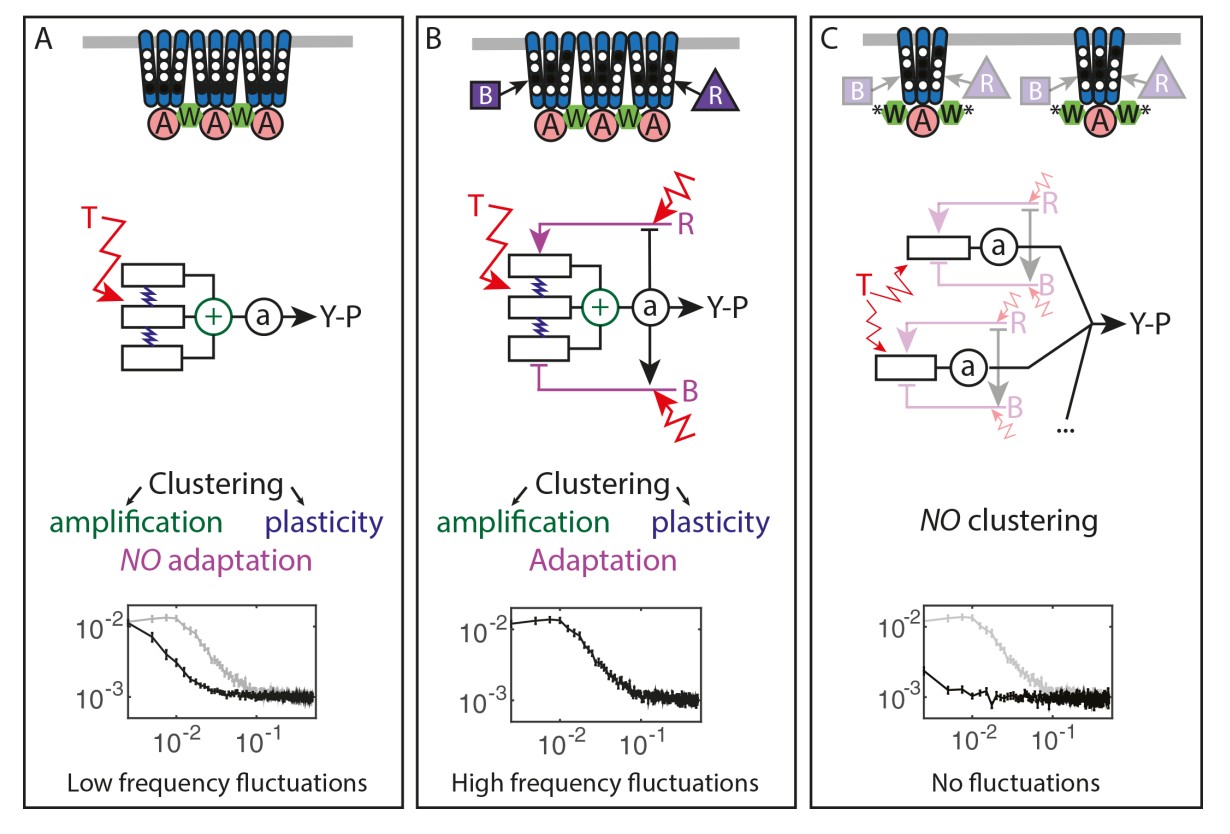

**Figure 5.** Multiple sources of signaling fluctuation in the chemotaxis pathway. (**A**) In the absence of adaptation enzymes, thermal fluctuations stimulating – and amplified by – the dynamic receptor cluster lead to low frequency fluctuations (<0.01 Hz) around intermediate cluster activity. The blue springs symbolize the plasticity of the receptor array, the green ⊕ its cooperativity. (**B**) In adapted wild-type cells, thermal fluctuations and fluctuations in the dynamics of CheR and CheB are amplified by the dynamic chemoreceptor cluster, which leads to fluctuations of the activity at frequencies around 0.03 Hz. (**C**) In the absence of clustering, responsive but non-amplifying receptor complexes do not produce observable activity fluctuations, whether or not adaptation enzymes are present. Graphs show the PSD of the FRET ratio measured in each respective case (black). In (**A,C**) the wild-type curve is shown for comparison (gray).

DOI: https://doi.org/10.7554/eLife.26796.026

less efficient neighborhood assistance by reducing the (de)methylation rate of the bound enzymes had only modest effects (*Figure 4—figure supplement 9C*).

## Discussion

Stochastic activity fluctuations are likely to have major impact on signal processing within cellular networks (*ten Wolde et al., 2016*; *Tsimring, 2014*). Nevertheless, direct visualization and characterization of such fluctuations at the posttranslational level remain limited to a small number of cases (*Conlon et al., 2016*; *Aoki et al., 2013*) primarily due to high requirements for the sensitivity and time resolution of the necessary single-cell measurements. Although fluctuations of the signaling activity can in some cases be deduced from the downstream output of the network, either gene expression (*Paliwal et al., 2007*; *Bowsher and Swain, 2012*) or behavior (*Korobkova et al., 2004*; *Emonet and Cluzel, 2008*; *Park et al., 2010*; *Pontius et al., 2013*), this output may strongly filter and reshape fluctuations. Consequently, the theoretical framework for the analysis of noise at the posttranslational level remains less developed than for variations in gene expression (*Raj and van Oudenaarden, 2008*; *Paulsson, 2004*).

Here we directly monitored activity fluctuations in the chemotaxis pathway of *E. coli*, a common model for quantitative analysis of signal transduction (*Tu, 2013*; *Sourjik and Wingreen, 2012*; *Micali and Endres, 2016*). One fascinating feature of the chemotaxis pathway is the amplification of

chemotactic signals through cooperative interactions within the clusters (arrays) of chemoreceptors, where at least ~10–20 receptor dimers show concerted transitions between active and inactive states (*Sourjik and Berg, 2004*; *Li and Weis, 2000*; *Duke and Bray, 1999*; *Mello and Tu, 2003*; *Monod et al., 1965*; *Keymer et al., 2006*). The pathway is also robust against external and internal perturbations, largely thanks to its methylation-based adaptation system (*Barkai and Leibler, 1997*; *Oleksiuk et al., 2011*; *Yi et al., 2000*; *Kollmann et al., 2005*; *Alon et al., 1999*). At the same time, the stochastic activity of the adaptation enzymes was also proposed as the reason for the observed strong variability in the signaling output, the duration of straight runs of the swimming cells (*Korobkova et al., 2004*; *Emonet and Cluzel, 2008*; *Pontius et al., 2013*). Indeed, inspired by so-called fluctuation-response theorems, previous analyses established a fluctuation-response relation between the adaptation time to stimuli (called response time) and the typical time scale of fluctuations of the tumbling rate in individual *E. coli* cells (*Emonet and Cluzel, 2008*; *Park et al., 2010*) – which we confirmed at the level of CheY phosphorylation ($\omega_{peak} \simeq \omega_{RB}$) – demonstrating that behavioral fluctuations originate within the chemotaxis pathway and pointing to the methylation system as their likely cause. Subsequently, the fluctuations in straight run durations were proposed to enhance environmental exploration, partly since the occasional long run allows exploring wider territories (*Emonet and Cluzel, 2008*; *He et al., 2016*; *Matthäus et al., 2009*; *Matthäus et al., 2011*; *Flores et al., 2012*).

Here we combined experimental and mathematical analyses to demonstrate that both, the adaptation system and receptor clustering contribute to the signaling noise in the chemotaxis pathway. Experimentally, we adapted the FRET-based assay that was previously applied to study average signaling properties in cell populations (*Shimizu et al., 2010*; *Sourjik and Berg, 2004*; *Endres et al., 2008*; *Frank and Vaknin, 2013*; *Piñas et al., 2016*; *Sourjik et al., 2007*; *Sourjik and Berg, 2002b*; *Oleksiuk et al., 2011*; *Neumann et al., 2014a*; *Krembel et al., 2015a*; *Meir et al., 2010*; *Frank et al., 2016*; *Krembel et al., 2015b*; *Neumann et al., 2014b*, Neumann et al., 2012*Neumann et al., 2012*; *Clausznitzer et al., 2010*), to be used at the single-cell level. Whereas previous studies have relied on the output provided by flagellar motor rotation (*Korobkova et al., 2004*; *Park et al., 2010*), using FRET enabled us to characterize the activity fluctuations directly, before their amplification by the motor. Our measurements showed that fluctuations can be comparable to the average adapted activity of the pathway and thus significantly larger than previous estimates (*Tu and Grinstein, 2005*). This surprisingly large amplitude of fluctuations indicates concerted variations of receptor activity across the signaling arrays containing hundreds to thousands of receptors. Furthermore, we showed that the stochasticity of receptor methylation could not be the sole cause of the pathway noise, because activity fluctuations were also observed in absence of the methylation system. In contrast, disruption of receptor clustering completely abolished these long-term activity fluctuations, even in presence of the methylation system, implying that receptor interactions are essential for the observed fluctuations.

To better understand the nature of the observed fluctuations, we applied analysis based on the fluctuation-dissipation theorem (FDT), following a recent theoretical study (*Sartori and Tu, 2015*). The FDT establishes a fundamental relationship between thermal fluctuations and the response to externally applied perturbations for an equilibrium system. Although being a powerful tool for studying equilibrium and out-of-equilibrium systems in physics (*Kubo, 1966*), so far it has found only limited application in biology (*Paulsson, 2004*; *Robert et al., 2010*; *Mizuno et al., 2007*; *Chevry et al., 2013*; *Bialek and Setayeshgar, 2005*). For the chemotaxis system, the FDT in its equilibrium form was used to predict the magnitude of thermally activated ligand binding noise with implications for maximal sensing accuracy (*Aquino et al., 2011*; *Bialek and Setayeshgar, 2005*). The present approach is also complementary to the previous fluctuation-response analysis mentioned above (*Emonet and Cluzel, 2008*; *Park et al., 2010*), itself conceptually related to the fluctuation theorems extending the FDT for certain systems in non-equilibrium steady states (*Park et al., 2010*; *Seifert, 2012*). Comparison of fluctuations and dissipation to evaluate whether the system deviates from the FDT, together with the analysis of mutants deficient in adaptation and/or clustering, enabled to identify multiple factors contributing to the pathway noise. These factors include (i) the input thermal noise, (ii) the amplification of this noise by cooperative interactions among receptors, (iii) the delayed response function of receptor clusters, and (iv) the dynamics of the methylation system (*Figure 5*).

Unexpectedly, the activity fluctuations in absence of the adaptation system could be explained for the most part by thermal noise acting on the receptors, which is amplified through the cooperative interactions of clustered receptors and subsequently converted into long-term pathway activity fluctuations by their slow response dynamics (*Figure 5A*). The contribution of out-of-equilibrium processes to these activity fluctuations seems to be minor if any. This phenomenon demonstrates that thermal noise can induce measurable fluctuations in activity of a cellular network, even in absence of active processes that are usually considered to be the main contributors to cellular dynamics. Even more striking is the amplitude of these fluctuations, suggesting that up to a half of the chemoreceptor array – that may contain thousands of receptors – flips its activity.

The slow cluster dynamics was recently observed using fluorescence anisotropy measurements and attributed to the stimulation-induced changes in packing of receptors within clusters (*Frank and Vaknin, 2013*). Indeed, in our experiments both slow response and activity fluctuations were abolished by mutations that disrupt clustering, suggesting that it corresponds to some large-scale plasticity within the receptor array (*Piñas et al., 2016*). Interestingly, such stimulation-induced slow reconfiguration had been also proposed to modulate cooperativity within the receptor array in an earlier theoretical study (*Hansen et al., 2010*). Although the precise mechanism behind this slow dynamics was not yet characterized, meaning that it could neither be experimentally disentangled from signal amplification nor mechanistically modeled, our simulations suggest that while slow dynamics sets the time scale of activity fluctuations, both this dynamics and amplification contribute to their amplitude. It thus seems that this previously little considered feature of the receptor array plays a large role in producing and shaping the activity fluctuations.

Our analysis also suggests that an effective subunit of the allosteric signaling teams corresponds to one trimer of dimers, rather than a dimer itself as assumed in previous computational models (*Endres et al., 2008*; *Mello and Tu, 2007*). This conclusion is consistent with several recent studies (*Piñas et al., 2016*; *Frank et al., 2016*; *Li and Hazelbauer, 2014*), and it could be easily reconciled with the previous formulations of the Monod-Wyman-Changeux models by rescaling the free-energy change per methylated glutamate by a factor of three. Since large size of the cooperative units implies fewer units per receptor array, it further helps to account for the large activity fluctuations even in absence of the methylation enzymes.

Notably, on the studied range of time scales the previously proposed contribution of the high-frequency ligand binding noise (*Aquino et al., 2011*; *Bialek and Setayeshgar, 2005*) to overall fluctuations must be very small, since the observed power spectral densities depended on activity but not on the absolute ligand concentration. The dynamics of CheY/CheZ interaction is also unlikely to contribute to the observed fluctuations because the turnover rate of this complex (>1 Hz) (*Li and Hazelbauer, 2004*; *Oleksiuk et al., 2011*) is above the frequency range of our experiments.

In the presence of the adaptation system the noise within receptor arrays is apparently added to the noise coming from the stochasticity of methylation events (*Figure 5B*), with both noise sources having comparable strength. The adaptation system not only shifts the frequency spectrum of fluctuations but also eliminates the latency of the response to stimuli, thus likely accelerating the response through its negative feedback activity. The statistics of methylation events inferred from the power spectra was compatible with previous understanding of the enzyme kinetics, including the hypothesis that methylation noise is enhanced by the ultrasensitivity to changes in the ratio of methylation enzymes (*Korobkova et al., 2004*; *Emonet and Cluzel, 2008*). Nevertheless, receptor clustering is required for the observed activity fluctuations even in presence of the adaptation system (*Figure 5C*), likely because of signal amplification as well as accelerated adaptation dynamics within clusters due notably to assistance neighborhoods (*Li and Hazelbauer, 2005*; *Pontius et al., 2013*; *Frank et al., 2016*). Our simulations suggested that the former likely plays a more prominent role in generating large activity fluctuations.

Altogether, the overall picture of the signaling noise in the chemotaxis pathway is more complex than previously suggested, with the noise being first processed through a slow responding amplifier (the chemoreceptor cluster) and then fed back through the methylation system, resulting in complex colored fluctuations of the pathway activity and therefore of the swimming behavior.

More generally, our study provides another example of the general relation between fluctuations and response in biological systems and it demonstrates that FDT-based analysis can distinguish between active and passive processes also within an intracellular network. Although activity fluctuations in biological systems are commonly shaped by active, out-of-equilibrium processes, meaning

that in many cases the FDT will not be satisfied (*Park et al., 2010*), the properties of a system can nevertheless be inferred when studying the deviation of its behavior from the FDT (*Robert et al., 2010*; *Martin et al., 2001*; *Mizuno et al., 2007*; *Chevry et al., 2013*). The approach of quantifying such deviations by means of an effective temperature, or fluctuation-dissipation ratio, has been used in a variety of out-of-equilibrium systems (*Cugliandolo, 2011*), from glasses to biological systems. Although in some systems, for example glasses, this ratio can have indeed properties normally associated with the thermodynamic temperature, in biological systems the effective temperature rather relates to the energy scale and frequency content of the underlying out-of-equilibrium processes. This relation was previously demonstrated for several systems, including the hair bundle of the inner ear (*Martin et al., 2001*) and active transport in eukaryotic cells (*Robert et al., 2010*; *Mizuno et al., 2007*; *Chevry et al., 2013*), and we show that it also applies to a signaling pathway. Notably, the present analysis differs both in its aims and technicalities from the aforementioned fluctuation-response analysis (*Emonet and Cluzel, 2008*; *Park et al., 2010*). For instance, the FDT breakdown in CheR$^+$CheB$^+$ cells does not contradict the previously observed relation between fluctuation and adaptation time scales, since these two observations provide different information: that the noise source encompasses an out-of-equilibrium process and that the fluctuations originate in the chemotaxis pathway, respectively. An interesting emergent feature of our analysis is the negative effective temperature, which arises as a hallmark of the delayed adaptive negative feedback (*Sartori and Tu, 2015*). A similar effect was also observed in inner ear hair bundles, where it is related to the mechanical adaptation feedback (*Martin et al., 2001*). Negative dissipation associated to the negative temperature was predicted to indicate a reversal of causality, induced here by adaptation (*Sartori and Tu, 2015*): Whereas positive dissipation means that changes in receptor free energy induce activity changes, negative dissipation results from the methylation system countering preceding activity changes (*Sartori and Tu, 2015*; *Sartori and Tu, 2011*; *Lan et al., 2012*) and actively translating them into free energy changes, thus opposing the passive behavior of the receptors. Importantly, because the FDT-based analysis requires only knowledge of system's fluctuations and its response, it is widely applicable for studying dynamics of diverse cell signaling processes, including those where molecular details are not known.

## Materials and methods

### Cell growth, media and sample preparation

*E. coli* strains and plasmids are listed in *Supplementary file 1A,B*, respectively. Cells carrying two plasmids that encode respectively Tar in the indicated modification states and the FRET pair were grown at 30°C overnight in tryptone broth (TB) supplemented with appropriate antibiotics. The culture was subsequently diluted 17:1000 in TB containing antibiotics, 2 μM salicylate (unless otherwise stated) for induction of Tar and 200 μM isopropyl β-D-1-thiogalactopyranoside (IPTG) for induction of the FRET pair, and grown at 34°C under vigorous shaking (275 rpm) to an OD$_{600}$ = 0.55. Bacteria were harvested by centrifugation, washed thrice in tethering buffer (10 mM KPO$_4$, 0.1 mM EDTA, 1 μM methionine, 10 mM lactic acid, pH 7) and stored at least 20 min at 4°C prior to the experiments.

### Microscopy

Bacterial cells were attached to poly-lysine coated slides which were subsequently fixed at the bottom of a custom-made, air-tight flow chamber, which enables a constant flow of fresh tethering buffer using a syringe pump (Pump 11 Elite, Harvard Apparatus, Holliston, Massachusetts, United States) at 0.5 ml/min. This flow was further used to stimulate cells with indicated concentrations of α-methyl-D,L-aspartate (MeAsp). The cells were observed at 40x magnification (NA = 0.95) using an automated inverted microscope (Nikon T*i* Eclipse, Nikon Instruments, Tokyo, Japan) controlled by the NIS-Elements AR software (Nikon Instruments). The cells were illuminated using a 436/20 nm filtered LED light (X-cite exacte, Lumen Dynamics, Mississauga, Canada), and images were continuously recorded at a rate of 1 frame per second in two spectral channels corresponding to CFP fluorescence (472/30 nm) and YFP fluorescence (554/23 nm) using an optosplit (OptoSplit II, CAIRN Research, Faversham, United Kingdom) and the Andor Ixon 897-X3 EM-CCD camera (Andor Technology, Belfast, UK) with EM Gain 300 and exposure time of 1 s (*Figure 1—figure supplement 1B*). For each measurement, the field of view was chosen to contain both a small region of high density

with confluent cells and a few hundred well-separated single cells (*Figure 1—figure supplement 1C*). During our approximately 30 min long measurements, the focus was maintained using the Nikon perfect focus system.

## Image processing and data analysis

The image analysis was performed using the NIS-Elements AR software. The CFP and YFP images, each recorded by a half of the camera chip ($256 \times 512$ px$^2$, 1 px = 0.40 µm), were aligned with each other by manual image registration. A gray average of the two channels was then delineated to enhance contrast and create binary masks with a user-defined, experiment-specific threshold. Individual cells were detected by segmentation of the thresholded image into individual objects, filtered according to size (3–50 µm$^2$) and shape (excentricity < 0.86). This step resulted in a collection of distinct regions of interest (ROIs) for each frame of the movie. The ROIs were then tracked from frame to frame, using the NIS build-in tracking algorithm. Only ROIs that could be tracked over the entire duration of the experiment were further analyzed. The selected ROIs were then inspected manually and those not representing individual single cells well attached to the cover glass were discarded. Each individual measurement contained on the order of 100 tracked single cells.

All further analyses were carried out using MATLAB 8.4 R2014b (The MathWorks, Inc., Natick, Massachusetts, United States). For each tracked cell, the average CFP and YFP values over the ROI were extracted as a function of time. These values were also extracted for an ROI corresponding to the confluent population of cells. The ratio $R$ of the YFP signal to the CFP signal was computed for both the single cells and the population, with the population response being used as a reference. Cells with a FRET ratio change of less than 10% of the population response were discarded as unresponsive. The PSD was computed over T = 400 frames long segments as

$$s_R(\omega) = \frac{1}{T} \left\langle \frac{\hat{R}_i(\omega)\hat{R}_i^*(\omega)}{\bar{R}_i^2} \right\rangle_i,$$

(4)

where $\hat{R}_i(\omega)$ is the discrete Fourier transform of the FRET ratio of cell $i$ at frequency $\omega/2\pi$, $\hat{R}_i^*$ its complex conjugate, $\bar{\phantom{x}}$ represents a temporal average over the given time interval and $\langle \cdot \rangle_i$ an average over all single cells considered. The error for the PSD was evaluated as $\frac{1}{N_c T} \mathrm{var}\left( \frac{\hat{R}_i(\omega)\hat{R}_i^*(\omega)}{\bar{R}_i^2} \right)_i$, where $N_c$ is the number of cells. The time autocorrelation function is simply the inverse Fourier transform of the PSD. The time autocorrelation functions were fitted by $C(t) = C_0 \exp(-t/\tau_0)$, for $t > 0$ to measure the correlation time $\tau_0$, $C_0$ being a free parameter accounting for the camera white shot noise. Although this fit was moderately accurate ($0.96 \leq R^2 \leq 0.98$ in all cases), it provided a simple estimate of the fluctuation time scale.

## Quantification of measurement noise

Contributions of technical fluctuations (vibrations, focus drift, *etc.*) and of the camera shot noise to the noise on the FRET ratio was quantified using fluorescent beads (BD FACSDiva CS and T Research beads #655050) that emit both in CFP and in YFP channels. The resulting shot noise was found to be perfectly white (*Figure 1—figure supplement 3A*). Additional negative control experiments were performed using a receptorless strain, where no CheA-based signaling occurs. In this case, the noise in FRET ratio was also mostly white, except at very low frequency (*Figure 1—figure supplement 3B*). Where indicated, the power spectra of other strains were corrected by subtracting the power spectrum of the receptorless strain, to obtain the 'pure' activity fluctuation spectra.

## Evaluation of the conversion factor $\lambda$

The value of $\lambda$, $0.10 \pm 0.01$, converting FRET ratio changes to kinase activity changes, was estimated using data for the $\Delta cheR\Delta cheB$ Tar$^{\mathrm{QEQE}}$ strain as $\lambda = \left\langle \bar{R}(0) \right\rangle - \left\langle \bar{R}(100\ \mu M) \right\rangle$, the difference, averaged over all cells, between the FRET ratio in buffer, where the activity should be maximal (i.e., equal to one), and the ratio upon saturating stimulation with 100 µM MeAsp. A similar value $\lambda = 0.09 \pm 0.01$ could be estimated in the adaptation-proficient strains, as the difference between the minimal FRET ratio value reached just after stimulation with 100 µM MeAsp and the maximal value reached upon removal of this stimulus. However, this latter value was slightly less precise because it

is not certain that full receptor activity is reached upon stimulation removal, and the more reliable $\Delta cheR\Delta cheB$ value was used in all cases.

## Activity sorting

For $Tar^{QEQE}$ receptors in non-adapting strains, we assumed that all the receptors are fully active in buffer conditions and fully inactive upon stimulation with 100 µM MeAsp. The pathway activity in each cell was thus evaluated as $A = 1 - \frac{\bar{R}(preStim-30\mu M) - \bar{R}(30\ \mu M)}{\bar{R}\ (preStim-100\ \mu M)\ -\bar{R}(100\ \mu M)}$. The use of the two different prestimulus values in buffer enables to minimize the effect of FRET baseline variation due to bleaching of fluorophores during image acquisition. Cells were then sorted according to their activity and divided into $n$ equally populated subpopulations, and for each subpopulation the average PSD $\langle s_R(\omega)\rangle_A$ at average activity $A$ of the subpopulation was evaluated for the set of frequencies displayed in *Figure 4—figure supplement 2*. This procedure was implemented for several values of $n$, namely $n = 10,\ 9,\ 6,\ 5\ \text{and}\ 4$, and the whole resulting data was used to plot $\langle s_R(\omega)\rangle_A$ as a function of $A$ (*Figure 4—figure supplement 2A*).

## Acknowledgements

The authors would like to thank R Somavanshi for assistance with experiments and NS Wingreen and SM Murray for comments on the manuscript.

## Additional information

### Funding

| Funder | Grant reference number | Author |
|---|---|---|
| European Research Council | 294761-MicRobE | Remy Colin<br>Christelle Rosazza<br>Victor Sourjik |
| Deutsche Forschungsgemeinschaft | German-Israeli Project Cooperation SO568/1-1 | Remy Colin<br>Victor Sourjik |
| Deutsche Forschungsgemeinschaft | German-Israeli Project Cooperation AM441/1-1 | Ady Vaknin |

The funders had no role in study design, data collection and interpretation, or the decision to submit the work for publication.

### Author contributions

Remy Colin, Conceptualization, Data curation, Software, Formal analysis, Investigation, Visualization, Methodology, Writing—original draft, Writing—review and editing; Christelle Rosazza, Data curation, Software, Formal analysis, Validation, Investigation, Visualization, Methodology, Writing—review and editing; Ady Vaknin, Resources, Funding acquisition, Investigation, Writing—review and editing; Victor Sourjik, Conceptualization, Formal analysis, Supervision, Funding acquisition, Validation, Methodology, Writing—original draft, Project administration, Writing—review and editing

### Author ORCIDs

Remy Colin (iD) http://orcid.org/0000-0001-9051-8003
Ady Vaknin (iD) http://orcid.org/0000-0002-4723-4600
Victor Sourjik (iD) http://orcid.org/0000-0003-1053-9192

### Decision letter and Author response

Decision letter https://doi.org/10.7554/eLife.26796.032
Author response https://doi.org/10.7554/eLife.26796.033

## Additional files

**Supplementary files**

• Supplementary file 1 List of strains and plasmids used in the study.
DOI: https://doi.org/10.7554/eLife.26796.027

• Transparent reporting form
DOI: https://doi.org/10.7554/eLife.26796.028

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

## Appendix 1

DOI: https://doi.org/10.7554/eLife.26796.029

This Appendix presents four partially independent theoretical derivations of equations and concepts presented in the main text.

# Modeling activity fluctuations in the framework of fluctuation-dissipation relation

## Fluctuation dissipation relation and effective temperature

In a system at equilibrium, a fluctuation-dissipation relation links the thermal fluctuations of any physical quantity to its response to an external small perturbation applied to the system via the temperature. It extends the corresponding fluctuation-response relation, which links the amplitudes of the fluctuation and the response, including their evolution in time. For the quantity $a$, it reads (**Kubo, 1966**):

$$-\frac{d}{dt}C_a(t,0) = kT\frac{\partial \Delta a(t)}{\partial \Delta h(0)},$$ (A1)

Where:

- $C_a(t,0) = \langle a(t)a(0)\rangle - \langle a(t)\rangle\langle a(0)\rangle$ is the time autocorrelation function of $a$,
- $h$ is the conjugate of $a$ in the Hamiltonian of the system, that is the Hamiltonian (*i.e.* the free energy) can be written $H = H_{unperturbed} - ah$
- $\frac{\partial \Delta a(t)}{\partial \Delta h(0)}$ is the response of $a$ at time $t > 0$ to the small impulse perturbation $\Delta h$ applied transiently at time 0. It is called impulse response function, usually denoted $\chi_a(t)$.

This relation can also be expressed in Fourier space (decomposing all temporal signals in terms of periodic functions) as: $s_a(\omega) = -\frac{2\,kT}{\omega}Im\,\hat{\chi}_a(\omega)$, where $s_a(\omega)$ is the power spectrum of $a$ and $\hat{\chi}_a(\omega)$ is the Fourier transform of $\chi_a(t)$, typically referred to as 'dynamic susceptibility'.

In a system which is not at equilibrium, but nonetheless at steady state, one can define the so-called fluctuation-dissipation ratio (**Cugliandolo, 2011**):

$$\frac{kT_{eff}(\omega)}{kT} = -\frac{\omega s_a(\omega)}{2kTIm\,\hat{\chi}_a(\omega)}.$$ (A2)

This FDT ratio is a way to quantify some ''distance to equilibrium'', introducing the effective temperature $T_{eff}(\omega)$. The system is in equilibrium only if the ratio equals one at all frequencies.

## Choice of model for the chemotaxis pathway

We model the receptors by two-state objects, being either kinase activating (ON) or kinase inhibiting (OFF). The free energy difference between ON and OFF is $f_0 = \gamma(m) + \eta(c)$ for a single receptor, with $\eta(c) = \ln\left(\left(1 + \frac{c}{K_{OFF}}\right)/\left(1 + \frac{c}{K_{ON}}\right)\right)$ being the contribution of attractant binding and $\gamma(m) = k_0 - k_1 m$ being the contribution of the receptor methylation.

Two models can describe the coupling between neighboring receptors and kinases in the chemoreceptor cluster, the Monod-Wyman-Changeux (MWC) and the Ising models (**Mello and Tu, 2003**; **Keymer et al., 2006**; **Mello and Tu, 2007**; **Skoge et al., 2006**). The MWC model considers that receptors are grouped by teams of $N_{MWC}$ infinitely coupled receptors and their associated kinases. The Hamiltonian of the whole chemoreceptor cluster is $H_{MWC} = \sum_{j=1}^{N_{team}}\left(a_j \sum_{k=0}^{N_{MWC}} \Delta f_0(k)\right)$, $a_j$ being the Boolean state of team $j$. The Ising model on the contrary considers finite coupling between receptors, and the Hamiltonian of the cluster is

$$H_{IM} = H_{int} + \sum_{k=1}^{N_T} a_k \, \Delta f_0(k), \tag{A3}$$

where $a_k$ is now the state of the single receptor dimer k and $H_{int}$ describes the coupling between and among receptors and kinases. The interaction term $H_{int}$ can be written in all generality

$$H_{int} = -J_{aa}\sum_{i,j}(A_i - 0.5)S_{i,j}(A_j - 0.5) - J_{ar}\sum_{i,k}(A_i - 0.5)V_{i,k}(a_k - 0.5) - J_{rr}\sum_{l,k}(a_l - 0.5)W_{l,k}(a_k - 0.5),$$

where $J$ are the coupling strengths, $S$, $V$ and $W$ describe the network by determining whether two components are coupled and $A_i$ is the Boolean activity of the *kinase i*.

**At steady state**, the average activity of the cluster is given in both cases by

$$A = \langle a \rangle = \frac{1}{Z}\int\int \frac{1}{N_T}\sum_{k=1}^{N_T} a_k \exp(-\beta H(\{a_k\})) \prod da_k, \tag{A4}$$

where $Z$ is a normalization factor, and $N_T$ is the total number of Tar dimers. In the MWC, it is solved exactly as $A_{MWC} = (1 + \exp(N_{MWC}\, f_0))^{-1}$. In the Is, analytical solutions exist only for a limited set of network topologies, but numerical solutions in most cases are well fitted by

$$A_{Is} = \frac{1}{1 + \exp(N\,\Delta f_0)}, \tag{A5}$$

where $N$ is a fitted parameter, corresponding to an effective "team size", which is proportional to the average number of neighboring receptors with the same activity (see 'Definition of the effective temperature').

The MWC does not allow individual receptors to fluctuate within their team nor any team rearrangement. This is unsatisfactory since individual receptors are expected to undergo independent thermal and/or active perturbations and the slow dynamics in the $\Delta cheR\Delta cheB$ strain might come from some remodeling of teams of receptors with the same activity (*Hansen et al., 2010*). The Ising model, which possesses those two properties, was therefore preferred.

Finally, the average methylation state of the receptor evolves under the action of CheR and CheB according to

$$\frac{dm}{dt} = k_R(1 - A) - k_B\, A, \tag{A6}$$

with $k_R$ and $k_B$ being the rates of methylation and de-methylation, respectively (*Clausznitzer et al., 2010*).

## Phenomenological step response function

To define the effective temperature (*Equation A2*), the activity state of a *single receptor dimer*, $a$, will be used as the variable. Considering the definition of the dynamic susceptibility (paragraph 1.1) and *Equation A3*, within the Ising model, the dynamic susceptibility $\chi_a(t)$ in response to a perturbation $+\epsilon$ of the free energy difference $\Delta f_0$ is

$$\langle \delta a(t) \rangle = \int_{-\infty}^{t} -\epsilon(\tau)\,\chi_a(t-\tau)d\tau, \tag{A7}$$

where $\langle \cdot \rangle$ is an ensemble average. In the case of a constant perturbation $\epsilon_0$ starting at $t = 0$,

$$\langle \delta a(t) \rangle = -\epsilon_0 \int_0^t \chi_a(\tau)\,d\tau. \tag{A8}$$

**In the absence of adaptation enzymes**, *Equation A5* implies that at steady state $\langle \delta a(+\infty) \rangle = -N\,\langle a \rangle\,(1 - \langle a \rangle)\,\epsilon_0$, which yields

$$\int_0^{+\infty} \chi_a^-(\tau)\, d\tau = N\langle a\rangle(1 - \langle a\rangle) \equiv \mathrm{X}_A^\infty. \tag{A9}$$

Here and in the following, we use a superscript '-' to refer to quantities in the $\Delta cheR\Delta cheB$ case, and superscript '+' for the CheR$^+$CheB$^+$ case.

In the all models so far, the activity switches very rapidly to its steady state value $\langle\delta a(+\infty)\rangle$, meaning that $\chi_a^-(\tau)$ is well approximated by a delta function. However, as observed in **Figure 4A** of the main text, step stimulation with MeAsp, which corresponds to the application of a constant $\epsilon_0$ to the receptors, induces also a long term dynamics of the activity, not captured by the models. A phenomenological description of this long term dynamics was therefore used, leading to a more complex form of $\chi_a^-(\tau)$.

We experimentally defined the step response function $g^-(t)$, measured as the response of a $\Delta cheR\Delta cheB$ strain to small step-like attractant stimulation, as

$$g^-(t) \equiv \frac{\langle\delta a(t)\rangle}{-\mathrm{X}_A^\infty \epsilon_0} = \frac{\langle\delta a(t)\rangle}{\langle\delta a(+\infty)\rangle} = \frac{\Delta R(t)}{-\lambda \mathrm{X}_A^\infty \epsilon_0} = \frac{\Delta R(t)}{\Delta R(+\infty)}, \tag{A10}$$

which goes from 0 at $t = 0$ to 1 at $t = +\infty$.

Combining **Equation A8, A9** and **A10**, the dynamic susceptibility of a receptor in the $\Delta cheR\Delta cheB$ strain is $\int_0^t \chi_a^-(\tau)\, d\tau = \mathrm{X}_A^\infty\, g^-(t)$, which is expressed in Fourier space as

$$\hat\chi_a^-(\omega) = \mathrm{X}_A^\infty\, i\omega\hat g^-(\omega) \tag{A11}$$

Here the Fourier transform of *x* is defined as

$$\hat x(\omega) = \int_{-\infty}^{+\infty} x(t)e^{-i\omega t}dt \tag{A12}$$

**In the CheR$^+$CheB$^+$ case**, by analogy we experimentally define the step response function to small step-like attractant stimulation as:

$$g^+(t) \equiv \frac{\langle\delta a^+(t)\rangle}{-\mathrm{X}_A^\infty \epsilon_0} = \frac{\Delta R^+(t)}{-\lambda \mathrm{X}_A^\infty \epsilon_0}, \tag{A13}$$

Here $\Delta R^+(t)$ is the measured YFP/CFP ratio during a small stimulation of free energy $\epsilon_0$ in CheR$^+$ CheB$^+$ cells expressing Tar only and $\lambda$ is the experimentally determined proportionality factor between FRET ratio and activity. Since the response is adaptive, the stimulation $-\lambda \mathrm{X}_A^\infty \epsilon_0$ cannot be deduced from the final change in FRET ratio ($\Delta R(+\infty)$). It was rather computed using $\langle A\rangle = 0.5$ and $\epsilon_0 = \ln(1 + \Delta c/K_{off})$, with $K_{off} = 7$ μM, which is lower than the value typically used for WT cells ($K_{off}=18$ μM) (**Kalinin et al., 2009**), to account for the increased sensitivity of the Tar-only strain at our expression level (**Krembel et al., 2015b**).

Similarly to the $\Delta cheR\Delta cheB$ case, the following relation holds:

$$\hat\chi_a^+(\omega) = \mathrm{X}_A^\infty\, i\omega\hat g^+(\omega) \tag{A14}$$

## Role of CheY/CheZ dynamics

In the previous sections, we have assumed that the concentration of CheY-P follows instantaneously the average activity of the cell. In practice, however, [CheY-P] is delayed compared to the activity. The CheY phosphorylation (by CheA)-dephosphorylation (by CheZ) cycle can be modeled by (**Vladimirov et al., 2008**):

$$\frac{dy}{dt} = \omega_{AY}A(y_{tot} - y) - \omega_{ZY}Zy \tag{A15}$$

In Fourier space, assuming that CheZ is abundant, the CheY-P perturbation $\delta y(\omega)$ follows the activity perturbation as (**Vladimirov et al., 2008**):

$$\delta y(\omega) = \delta y_{max} \frac{\omega_Y}{\omega_Y + i\omega} \delta a(\omega) \tag{A16}$$

The characteristic frequency is $\omega_Y = 2$Hz (**Vladimirov et al., 2008**), which lies in the range of frequencies for which our measurements are dominated by instrumental noise. Therefore CheY/CheZ dynamics was neglected.

## Effect of diffusive smoothing of the step function

We assumed a step increase of the attractant concentration when measuring the response functions. In practice, because of mixing while delivering the media to the cells, the attractant concentration step is smoothed and it takes about 1 s to reach maximal concentration. We note $\epsilon(t) = \epsilon_0 \epsilon_s(t)$ the actual experimental free energy change experienced by the cells, with $\epsilon_s(t)$ a function which is zero for $t<0$, and rise to one in a time scale of the order of 1 s. Typically, $\epsilon_s(t) = 1 - \exp(-t/\tau_s)$ with $\tau_s = 0.5$ s. In Fourier space, the actually measured activity change is

$$\langle \delta a_{meas}(\omega) \rangle = -X_A^\infty \epsilon_0 \ \frac{\chi_a(\omega)}{X_A^\infty} \epsilon_s(\omega) \tag{A17}$$

**Equation A7, A10, A13** and **Equation A17** yield a relation between the actually measured response functions $g_{meas}^\pm(\omega)$ and their ideal counterpart $g_{ideal}^\pm(\omega)$– if the perturbation were purely step-like:

$$\frac{\langle \delta a_{meas}(t) \rangle}{-X_A^\infty \epsilon_0} \equiv g_{meas}^\pm(\omega) = i\omega\epsilon_s(\omega) \ g_{ideal}^\pm(\omega) \tag{A18}$$

For the typical exponential perturbation, when $\omega \neq 0$

$$i\omega\epsilon_s(\omega) = 1 - \frac{i\omega\tau_s}{1 + i\omega\tau_s} \tag{A19}$$

**Equation A19** reduces to one in the range of frequencies for which our measurement is above noise, and $g_{meas}^\pm(\omega) = g_{ideal}^\pm(\omega)$ was assumed for most of the analysis. Only for measuring the time scale of adaptation $g^+$ and the relation between $g^-$ and ('Effect of diffusive smoothing of the step function') of this supplement) was the full **Equation A18** needed.

## Definition of the effective temperature

**Equation A2** and **Equation A11** or **Equation A14** lead to:

$$\frac{kT_{\text{eff}}(\omega)}{kT} = -\frac{s_a(\omega)}{2 X_A^\infty Re(\hat{g}(\omega))}, \tag{A20}$$

Thus, to compute the effective temperature, we need to evaluate the power spectral density (PSD) of the activity of a single receptor dimer, $s_a(\omega)$. We experimentally have access to the PSD of the YFP/CFP ratio, the fluctuations of which are proportional to the ones of the average activity of the cell $A_{cell}$ with the factor $\lambda$, modulo the camera noise, so that

$$s_R(\omega) = \lambda^2 \ s_{A_{cell}}(\omega) + \ \epsilon_n^2. \tag{A21}$$

The average activity of the cell is given by $A_{cell} = \frac{1}{N_T} \sum_{k=1}^{N_T} a_k$, so that

$$s_{A_{cell}}(\omega) = \frac{1}{TN_T^2} \sum_{k=1}^{N_T} \sum_{k'=1}^{N_T} \langle \delta a_k(\omega) \delta a_{k'}^*(\omega) \rangle, \tag{A22}$$

Since receptors are coupled, $\langle \delta a_k(\omega) \ \delta a_{k'}(\omega) \rangle$ is not necessarily zero. In the Ising model, we have $\langle \delta a_k(\omega) \delta a_{k'}^*(\omega) \rangle = \left\langle |\delta a_k(\omega)|^2 \right\rangle C(r_{kk'})$, where $C(r_{kk'})$ is the correlation function between receptors distant from $r_{kk'}$ on the lattice, which decreases exponentially on a given length

scale (**Berg, 2003**), so that $\sum_{k=1}^{N_T} \sum_{k'=1}^{N_T} \left\langle \delta a_k(\omega) \delta a_{k'}^*(\omega) \right\rangle = N_T N_r \left\langle |\delta a_k(\omega)|^2 \right\rangle$, where $N_r$ is the average number of correlated receptors in the cluster (the loose equivalent of the team size of the MWC model), which is expected to be proportional to the cooperativity number N (**Equation A5**).

To accurately count the number of correlated receptors, we noted that recent works measured in vitro (**Li and Hazelbauer, 2014**) and in vivo (**Piñas et al., 2016**; **Frank et al., 2016**) the response function of the minimal functional chemosensory assembly, believed to consist of two trimers of receptor dimers (TD) coupled to one CheA dimer, and found a cooperativity number close to 2. The dose-response curve of $\Delta cheR \Delta cheB$ CheW-X2 expressing Tar$^{\text{QEQE}}$, featuring such minimal complexes (**Piñas et al., 2016**; **Frank et al., 2016**), was fitted using **Equation A5** (**Figure 3—figure supplement 1**), also yielding $N \simeq 2$. These results strongly suggest that N effectively accounts for the number of TDs coupled in a 'signaling team', thus $N_r = 3N$ and:

$$s_{A_{cell}}(\omega) = \frac{3N}{N_T} s_a(\omega) \tag{A23}$$

Finally, **Equation A9**, **A20**, **A21** and **A23** yield:

$$\frac{k \mathrm{T}_{\text{eff}}(\omega)}{k \mathrm{T}} = -\frac{N_T}{6 \lambda^2 N^2 A (1-A)} \frac{s_R(\omega) - \epsilon_n^2}{Re(\hat{g}(\omega))}, \tag{A24}$$

corresponding to **Equation 1 and 2** of the main text, which defines the dissipation $G_R(\omega) = -2 \lambda^2 \frac{3N^2 A (1-A)}{N_T} \mathrm{Re}(\hat{g}(\omega))$.

Note that although we expressed the fluctuation dissipation relation in terms of activity, which allows us to directly compare the analysis with experimental data, this relation can be formulated for any variable (e.g., receptor conformation) that itself determines the activity.

## Link between the response functions in $\Delta cheR \Delta cheB$ and CheR$^+$ CheB$^+$ cases

In presence of the adaptation system, the receptor cluster is assumed to respond to free energy perturbations in the same way as in the adaptation-deficient cells, but this response induces a methylation change adding up to the free energy perturbation. In Fourier space, for a small perturbation of the free energy difference $\epsilon(\omega)$, the resulting perturbations for the average activity and methylation are then given – from **Equation A6** and **Equation A7** – by the set of equations:

$$\langle \delta a^+(\omega) \rangle = \mathrm{X}_A^\infty \; i\omega \hat{g}^-(\omega) \; (-\epsilon(\omega) + k_1 \langle \delta m(\omega) \rangle) \tag{A25}$$

$$i\omega \langle \delta m \rangle = -(k_R + k_B) \langle \delta a^+(\omega) \rangle \tag{A26}$$

Defining $\omega_{RB} = \mathrm{X}_A^\infty k_1 (k_R + k_B)$, the activity dependent rate of adaptation, this set of equations is easily solved as

$$\langle \delta_a^+(\omega) \rangle = \frac{\mathrm{X}_A^\infty \, i\omega \, \hat{g}^-(\omega)}{1 + \omega_{RB} \, \hat{g}^-(\omega)} (-\varepsilon(\omega)) \tag{A27}$$

We thus inferred the dynamic susceptibility in CheR$^+$ CheB$^+$ as

$$\hat{\chi}_a^+(\omega) = \frac{\mathrm{X}_A^\infty \, i\omega \, \hat{g}^-(\omega)}{1 + \omega_{RB} \, \hat{g}^-(\omega)} \tag{A28}$$

Note that the $\Delta cheR \Delta cheB$ case is obtained again if $\omega_{RB} = 0$.

From **Equation A14**, the step response functions in the CheR$^+$ CheB$^+$ and $\Delta cheR \Delta cheB$ cases are linked by:

$$\hat{g}^+(\omega) = \frac{\hat{g}^-(\omega)}{1 + \omega_{RB}\,\hat{g}^-(\omega)} \tag{A29}$$

## Effect of diffusive smoothing of the step function

In the case where the stimulation is not a perfect step function, modeled by $\epsilon(\omega) = \epsilon_0 \epsilon_s(\omega)$, using **Equation A17** and **Equation A18**, the relation of equivalence can be easily shown to become:

$$\hat{g}^+_{meas}(\omega) = \frac{i\omega\epsilon_s(\omega)\ \hat{g}^-_{meas}(\omega)}{i\omega\epsilon_s(\omega) + \omega_{RB}\,\hat{g}^-_{meas}(\omega)}. \tag{A30}$$

Using $\epsilon_s(t) = 1 - \exp\left(-\frac{t}{\tau_s}\right)$, with $\tau_s = 0.5$ s, the equivalent of **Equation A30** in real space was fitted using the experimentally determined $\hat{g}^-_{meas}(\omega)$ and $\omega_{RB}$, with $\hat{g}^+_{meas}(\omega)$ as a free parameter, yielding $\omega_{RB} = 0.06$ Hz (**Figure 4—figure supplement 5**).

## Frequency of effective temperature divergence

In the CheR$^+$ CheB$^+$ case, the effective temperature diverges when $\mathrm{Re}\,\hat{g}^+(\omega) = 0$. **Equation A29** thus yield an implicit equation for the frequency at which this divergence occurs, $-\mathrm{Re}\,\hat{g}^-\left(\omega_{dvg}\right) = \omega_{RB}\left|\hat{g}^-\left(\omega_{dvg}\right)\right|^2$, which has a solution since $\mathrm{Re}\,\hat{g}^-(\omega)$ is negative. This equation clearly represents a balance between the action of the cluster cooperative response (represented by $g^-$) and adaptation (represented by $\omega_{RB}$). The solution is however not trivial, in particular $\omega_{dvg} \neq \omega_{RB}$, and will depend on both the time scales of cluster dynamics and adaptation. Notably, in (**Sartori and Tu, 2015**) the typical time scale of the cluster dynamics was chosen to be much shorter than the one suggested by our measurements, resulting in higher frequency of effective temperature divergence.

## Separating the contribution of methylation enzymes dynamics to the PSD in CheR$^+$ CheB$^+$ cells

A complementary approach to the modeling of the fluctuating activity of chemoreceptor clusters, which has been used in a number of previous theoretical works (**Clausznitzer and Endres, 2011**; **Sartori and Tu, 2011**; **Aquino et al., 2011**), is to introduce noise terms in **Equation A25** and **Equation A26**, which describe the average behavior of the system, in order to describe the behavior of single receptor $k$:

$$\delta a_k(\omega) = \mathrm{X}_A^\infty\ i\omega\hat{g}^-(\omega)\ (-\epsilon_k(\omega) + k_1\delta m_k(\omega)) \tag{A31}$$

$$i\omega\,\delta m_k = -(k_R + k_B)\delta a_k(\omega) + \delta r_k(\omega) + \delta b_k(\omega) \tag{A32}$$

Here $\epsilon_k(\omega)$ represents thermal noise acting on the receptor, and $\delta r_k(\omega)$ and $\delta b_k(\omega)$ represent noise coming from the intermittent action of CheR and CheB, respectively (see below for possible interpretation of these fluctuations).

This set of equations is easily solved as

$$\delta a_k(\omega) = \mathrm{X}_A^\infty \hat{g}^+(\omega)(-i\omega\epsilon_k(\omega) + k_1(\delta r_k + \delta b_k)), \tag{A33}$$

where $\hat{g}^+(\omega)$ is defined by **Equation A29**, and can be measured using **Equation A13**. Assuming that the power spectra of $\delta r_k$ and $\delta b_k$ are identical, denoted $s_{rb}(\omega)$, the power spectrum of the activity of one receptor is:

$$s_a^+(\omega) = \left|\mathrm{X}_A^\infty \hat{g}^+(\omega)\right|^2 \left(\omega^2 s_\epsilon(\omega) + 2k_1^2 s_{rb}(\omega)\right). \tag{A34}$$

This equation highlights the contributions of thermal fluctuations and methylation noise to the PSD. If the methylation system is absent, this latter equation reduces to the $\Delta cheR\Delta cheB$ case:

$$s_a^-(\omega) = \left|X_A^\infty \hat{g}^-(\omega)\right|^2 \omega^2 s_\epsilon(\omega). \tag{A35}$$

Under the non-trivial assumption that the thermal noise term (which can be explicitly evaluated using the FDT, **Equation A20**) remains the same whether adaptation enzymes are present or not, the contribution of the enzymes to the PSD in CheR$^+$ CheB$^+$ is:

$$s_a^m(\omega) = s_a^+(\omega) - \left|\frac{g^+(\omega)}{g^-(\omega)}\right|^2 s_a^-(\omega) = \left|k_1 X_A^\infty \hat{g}^+(\omega)\right|^2 s_{rb}(\omega), \tag{A36}$$

which yields in terms of the FRET ratio, from **Equation A21** and **Equation A23**:

$$s_R^m(\omega) = s_R^+(\omega) - \left|\frac{g^+(\omega)}{g^-(\omega)}\right|^2 s_R^-(\omega) = \frac{3N\lambda^2}{N_T}\left|k_1 X_A^\infty \hat{g}^+(\omega)\right|^2 s_{rb}(\omega). \tag{A37}$$

Here the thermal noise contribution in presence of adaptation is $s_R^T(\omega) = \left|\frac{g^+(\omega)}{g^-(\omega)}\right|^2 s_R^-(\omega)$.

## Possible interpretation of the methylation-based noise term

The non-perturbative equation for the evolution of the methylation of receptor $k$ reads:

$$\frac{dm_k}{dt} = - w_b b_k a_k + w_r r_k (1 - a_k) \tag{A38}$$

Here $b_k$ and $r_k$ evaluate whether, respectively, CheB or CheR is present on the site to act on the receptor, with respective rates $w_b$ and $w_r$, in an activity-dependent manner. This equation accounts for the fact that CheR and CheB, which are in low amounts compared to the total amount of receptors, bind and unbind in the vicinity of only a given number of receptors. Hence not all receptors are (de)methylated at a given time (**Pontius et al., 2013**). The ensemble average of **Equation A38**, describing the average methylation dynamics, is:

$$\frac{d\langle m\rangle}{dt} = - w_b \frac{N_B}{N_T} \langle A\rangle + w_r \frac{N_R}{N_T} (1 - \langle A\rangle) \tag{A39}$$

This identifies the ensemble averaged rate of (de)methylation, $k_R = w_r \frac{N_R}{N_T}$ ($k_B = w_b \frac{N_B}{N_T}$). Subtracting **Equation A39** from **Equation A38** leads to the perturbative **Equation A32**. This enables to define $\delta b_k$ and $\delta r_k$ as:

$$\delta r_k = w_r (1 - \langle A\rangle)\left(r_k - \frac{N_R}{N_T}\right) \tag{A40}$$

$$\delta b_k = w_b \langle A\rangle\left(b_k - \frac{N_B}{N_T}\right) \tag{A41}$$

These equations enable to identify $\delta r_k$ ($\delta b_k$) as the fluctuations in occupancy of a given receptor by CheR (CheB) and thus $s_{rb}(\omega)$ as the power spectrum of enzyme binding dynamics.

Although noisy, $s_{rb}(\omega)$ appeared to decrease at low frequency (**Figure 4—figure supplement 8**). Such a decrease indicates anti-correlations (**Peng et al., 1993**) in the binding dynamics of the methylation enzymes at their substrates, which is consistent with the common assumption that CheR (CheB) loads and acts only on the inactive (active) receptor. For the example of CheR, this activity dependence implies that once receptor is active, it will not allow CheR to reload and restart acting until it switches back into the inactive state, thus introducing a delay in the rebinding of the enzyme. As a consequence, enzyme binding anti-correlates on the time scale of this delay.

# Simulation of a simplified model for the array of receptors

In order to reproduce semi-qualitatively the features of the CheR$^+$ CheB$^+$ behavior displayed in **Figure 4**, a simple model of the receptor array was simulated. The standard values of all simulation parameters are given in **Appendix 1–table 1**. The simulated array is composed of $N_{\text{team}} = 300$ independent MWC signaling teams. The MWC model was chosen for simplicity, and it is expected to lead to qualitative but not necessarily quantitative match between simulations and experiments. Each signaling team is composed of $N_{rcp} = 3N$ receptor dimers – each of which counts eight methylation sites. The Boolean activity $a_k$ of the signaling team evolves according to:

$$\frac{da_k}{dt} = -w_a\left(a_k - \frac{1}{1+e^F}\right), \; F = N\,\Delta f_0 - k_1\left(m_k - N_{rcp}\,m_0\right) \tag{A42}$$

Here, $\Delta f_0$ is the attractant dependant stimulation, $w_a$ is the flipping rate of the kinase and $m_k$ is the total methylation level of the team.

**Appendix 1—table 1.** Parameters used in the simulations.

| Parameter | Value | Reference |
|---|---|---|
| $N$ | 14 | This study (based on experimental values) |
| $N_{team}$ | 300 | This study (based on experimental values) |
| $k_1$ | 0.016 | Adapted from (**Clausznitzer et al., 2010**) |
| $w_m$ | 1 s$^{-1}$ | This study |
| $w_l$ | 0.15 s$^{-1}$ | This study |
| $w_u$ | 0.5 s$^{-1}$ | This study |
| $w_y$ | 1 s$^{-1}$ | (**Vladimirov et al., 2008**) |
| $w_a$ | 0.25 s$^{-1}$ | This study |
| $b_{tot}$ | 240 | (**Li and Hazelbauer, 2004**) |
| $r_{tot}$ | 140 | (**Li and Hazelbauer, 2004**) |

DOI: https://doi.org/10.7554/eLife.26796.030

If $m_k$ is fixed, **Equation A42** is a simple model for the $\Delta cheR\Delta cheB$ case. Since the activity of a single team can only take 0 or 1 as a value, it fluctuates between these two values, being only on average equal to $1/(1+e^F)$. Since the teams are uncoordinated, the average activity of the whole cluster will fluctuate as well. This dynamics represents the thermal fluctuations in a MWC model. This dynamics was simulated for T = 1000 s after an equilibration period from a random initial condition of same duration, for $n = 100$ repeats, with $F = 0$, that is $\langle a \rangle = 0.5$. Increasing latencies in the response to stimulations of the receptor cluster were modeled by decreasing $w_a$, for a fixed amplification factor $N = 14$. As expected, the thermal fluctuations were slower for lower $w_a$. The maximal amplitude of the fluctuations was also larger when $w_a$ was larger (**Figure 4—figure supplement 4**). Increasing N while keeping the total number of receptors constant (*i.e.* decreasing $N_{team}$ accordingly), at fixed $w_a$, led to an increased amplitude of the fluctuations, their temporal dependences being however not affected (**Figure 4—figure supplement 4**). The power spectra however differed from experimental data. The amplitude was underestimated because the MWC does not allow applying thermal fluctuations to individual receptor. The time dependence was also different because we modeled the slow receptor cluster dynamics by lengthening the switching rate $w_a$, which is the only time scale of the model, where in reality they probably are different processes.

In the CheR$^+$ CheB$^+$ case, the methylation level evolves according to:

$$\frac{dm_k}{dt} = w_m \left( r_k \,\&\, (m_k{<}8N_{rcp}) - b_k \,\&\, (m_k{>}0) \right) \qquad (A43)$$

Here $r_k$ ($b_k$) represents whether a CheR (CheB) protein is tethered to the team. Importantly, the model assumes that only one enzyme may be tethered to the team at a time. (De)methylation occurs at the rate $w_m$ if CheR (CheB) is present and $m_k$ has not reached its maximal (minimal) value. The enzyme tethering dynamics is given by the set of equation:

$$\frac{dr_k}{dt} = w_l(1-r_k)(1-b_k)(1-a_k) - w_u r_k a_k \qquad (A44)$$

$$\frac{db_k}{dt} = w_l(1-r_k)(1-b_k)a_k - w_u b_k(1-a_k) \qquad (A45)$$

under the constraint $\sum_k b_k \leq b_{tot}$ and $\sum_k r_k \leq r_{tot}$. This means that CheR (CheB) may only load, if free enzymes are available, on free inactive (active) receptors with rate $w_l$ and unload once the receptor turned active (inactive) with rate $w_u$.

The typical dynamics in the simulation will then be the following. Take, for example, a weakly methylated team. Its activity will get to zero (**Equation A42**). If free CheR is available, it will load on the team (**Equation A44**) and methylate it (**Equation A43**), until the methylation level is high enough to activate the team (**Equation A42**). CheR will then unload (**Equation A44**), and a hypothetic free CheB can then load on the team (**Equation A45**) to demethylate it and bring it to its initial state.

The level of phosphorylated CheY of the simulated cell, also used as an output of the model, evolves according to:

$$\frac{dy}{dt} = w_y \left( \frac{1}{N_{team}} \sum_{k=1}^{N_{team}} a_k - y \right) \qquad (A46)$$

Output quantities were averaged over $n = 100$ independent simulations of single cells.

In practice, $w_u$, $w_l$, and $w_a$ were chosen of the same order of magnitude, and they were the slowest dynamics, whereas $w_m$ was the fastest, in order to obtain reasonable dynamics.

Starting from a random initial condition, the system was let to equilibrate at $\Delta f_0 = 0$ for 100 times the slowest time scale of the system ($1/w_l$). The system was then challenged with free energy perturbation $\Delta f_0 = \ln(1 + 0.3/7)$ (mimicking the experimental conditions) to measure the step response function, computed as $g_{simu}(t) = \frac{\Delta y(t)}{N_{eff}\Delta f_0}$. **Figure 4—figure supplement 9A** shows the normalized step response function compared to its experimental counterpart with excellent agreement (although absolute amplitudes differed moderately).

A $T = 800$ s equilibrated run was further used to compute power spectra, using **Equation A5** of the main text. The power spectra of $r_k$ and $b_k$, corresponding to the inferred $s_{rb}(\omega)$ defined in **Equation A36**, show good qualitative agreement with the experimental data, with a transition from high values at frequencies larger than 0.01 Hz to low values below this threshold (**Figure 4—figure supplement 9B**). This transition indicates anti-correlations in the occupancy of the receptor teams by the enzymes, which emerge from their *activity-dependant* loading and unloading. The two spectra are equal within noise by construction of the model (r and b play symmetric roles). Furthermore, the simulated power spectrum of the activity $s_A(\omega)$ was similar to the experimental power spectrum corrected for long term cluster dynamics (compare **Figure 4D** with **Figure 4—figure supplement 9C**). The amplitude of the power spectrum was however ~ 100 fold lower than in experiments, but in line with previous simulations (**Sartori and Tu, 2015**).

Finally, from the power spectrum of the CheY-P level $s_Y(\omega)$, which was very close to $s_A(\omega)$, an effective temperature can be computed as

$$\frac{\mathrm{T}}{\mathrm{T_{eff}}} = \frac{2\langle \mathrm{A} \rangle (1 - \langle \mathrm{A} \rangle)}{N_{team}} \frac{\hat{g}_{simu}(\omega)}{s_Y(\omega)} \qquad (A47)$$

It compares qualitatively well with the experimental effective temperature, with concordant frequencies of divergence (*Figure 4—figure supplement 9D*). Differences appear for the lowest frequencies, probably because of the long-term dynamics of the receptor clusters, which was not accounted for by these simulations.

All things being otherwise equal, modifying $N$ to 2 and $N_{team}$ to 2100, which models the disruption of the chemoreceptor clusters into individual trimers of dimers, reduced strongly the fluctuations in activity (*Figure 4—figure supplement 9C*). Decreasing the specific rate of receptor (de)methylation when the enzyme is bound to the receptors, $w_m$, to $w_m = 0.016$ s$^{-1}$ had however little effect (*Figure 4—figure supplement 9C*). Note that in both cases the adaptation time is reduced by a similar factor (7 and 6, respectively), since this time is proportional to the product $N\omega_m$, as evident from (*Equation A41*) and (*A42*).

Conditional tethering of the adaptation enzymes to the receptors therefore seems to account relatively well for the observed dynamics. One important discrepancy between simulations and experiments is in the amplitudes of the fluctuations, which are much larger than expected in experiments, when the simple MWC model is considered.

