## [Decision Letter]

Thank you for submitting your article "Multiple sources of signaling noise in bacterial chemotaxis network" for consideration by *eLife*. Your article has been reviewed by three peer reviewers, and the evaluation has been overseen by Naama Barkai as the Senior and Reviewing Editor. The reviewers have opted to remain anonymous.

The reviewers have discussed the reviews with one another and the Reviewing Editor has drafted this decision to help you prepare a revised submission.

Summary:

FRET technique is applied to study the contributions to demonstrate that variability in the methylation or cooperativity of the chemotactic receptor to contribute to signaling noise in the bacterial chemotaxis system. This extends previous studies which focused on gene-regulatory noise. To this end, the authors analyze mutants that are defective in methylation or in clustering. A theoretical framework, based on fluctuation-dissipation theorem (FDT), is used to interpret the data.

Essential revisions:

As you can see from the individual reviews, all three reviewers found the subject interesting and that the work has the potential to be of broad interest. However, some significant revisions are still required.

Please answer in detail all of the requests made by the reviewers. In particular, the following points were emphasized during the discussion:

1) Presentation should be improved in terms of motivation, schematics, and readability.

2) Additional controls are required more make sure that noise in single-cell FRET is really due to processes inside cells, e.g. by imaging background without cells (receptor-less strain not enough).

3) More rigor is needed in the theory development (best in an appendix), putting in context of previous FDT work in bacterial chemotaxis, and stressing novelty.

4) The main novelty is noise due to cooperative receptors, which is discussed only very briefly in paper. This claim would potentially require more elaboration (e.g. to separate receptor rearrangement from signaling noise). For the latter, receptor could be overexpressed or crosslinked to increase clustering. Theory is vague at best including Ising vs. MWC, 'adjustment' and 'rescaling' of data, and non-constant effective Temperature, and limits of applicability of FDT also need to be clarified.

*Reviewer #1:*

The aim of this paper is to experimentally demonstrate multiple sources of signaling noise in the bacterial chemotaxis pathway, as most previous works dealt with gene-regulatory noise. Using the in vivo FRET technique in the bacterium *E. coli*, the authors show that there are significant contributions to the noise from receptor methylation (shown previously by the Cluzel lab) as well as receptor cooperativity. For the former, the authors use receptor mutants in fixed methylation-like states without the adaptation enzymes CheR and CheB. For the latter they utilize the CheW-Χ2 mutant, where the adapter protein CheW does not lead to receptor clustering (but nevertheless signals to the downstream kinase). As a control, they also exploit cells without receptors to estimate the noise floor in absence of receptor signaling and adaptation. The results are cast in the form of the fluctuation-dissipation theorem (FDT), which, for near-equilibrium systems, says that the power spectrum of the equilibrium fluctuations equals the response to a (small) nonequilibrium perturbation. In practice, the FDT is often used to predict nonequilibrium behavior using only knowledge from equilibrium fluctuations. How this might apply to nonequilibrium systems is still subject of ongoing research. However, here the authors use the concept of an effective temperature to frame the nonequilibrium problem as an equilibrium-like problem. The negative sign of the effective temperature is interpreted as evidence for a strongly driven, nonequilibrium system. The topic of the paper is of general interest to people working in quantitative biology. However, there are a number of concerns regarding presentation of the results and derivation of the theory, which would need to be addressed thoroughly.

Major experimental issues:

Subsection “Role of receptor cooperativity in signaling noise”: The noise from receptor methylation has been investigated before [Emonet and Cluzel, 2008; Park et al., 2010], so that the main experimental novelty is the signaling noise from receptor cooperativity. What is the nature of this noise – fluctuations in the signaling state or rearrangement of receptor clusters? Is there any way to separate these? This novelty section is only 10 lines long.

Subsection “Fluctuation dissipation relation for receptor clusters” on FDT: There has been significant amount of work on FDT-like properties including [Emonet and Cluzel, 2008] and [Park et al., 2010]. In the latter, they talk about the fluctuation-response theorem. What is the difference, and how does the current work fit into the context of the previous works?

Figure 4 legend: The caption says that "adaptation deficient" cells were used to obtain response function in Figure 4. But the response function shown in this panel overshoots and adapts. How can these be the adaptation-deficient cells?

Figure 4: These panels show power spectra and response functions for both the non-adapting (B) and adapting (C) strains, but panel A only shows response function for one strain. Where is the other response function? Panel C shows three curves but legend only mentions two-line styles. What does "rescaled" in B and "adjustment" in C really mean (in plain English)? To obtain response function, a 30 μM stimulus was apparently used, but is result really independent of stimulus strength (as long as small), or does each case need to be adjusted or rescaled separately? The theory in Materials and methods seems to be developed used an oscillating stimulus. Does it matter what the stimulus looks like?

Figure 4: For the FDT to make any sense, T_eff_ should be constant (independent of omega), which seems to be the case when plotted as in panel D. But upon closer inspection, the blue curve changes from 2 to almost zero, so this isn't constant after all.

Major theoretical issues:

Introduction, fifth paragraph and subsection “Modeling activity fluctuations in the framework of fluctuation dissipation relation”, first two paragraphs (Materials and methods): both the Ising and the MWC model are used. Why both? Why not just use the simpler MWC model?

Subsection “Fluctuation dissipation relation for receptor clusters” on FDT: A major issue is that the theory is only described very briefly in the Materials and methods section, and that it is almost impossible to understand. Why is it not derived in detail, e.g. in the Appendix (which is currently only one page long). When looking through the literature, found paper Clausnitzer and Enders, BMC SB (2011), which also seems to derive response functions and signaling noises. How does the current work relate to their theory and predictions?

Another theory paper, [Sartori and Tu, 2015], says that the FDT breaks down or is violated in the chemotaxis pathway. Why is the FDT then used to study the chemotaxis pathway? Why do power spectra and response functions even match? Maybe the rescaling/adjustment really means that they don't match.

*Reviewer #2:*

This paper aims at quantifying beside the methylation/adaptive system whether receptors can contribute to the signaling noise of the chemotaxis network in *E. coli*. The authors use a robust technique based on the tandem FRET CheY-YFP and CheZ-CFP that has been well-established about more than 10 years ago. The technical improvement from older studies by the same group is that they now use a CCD camera to monitor fluctuations in the Fret signal at the single cell level. This single-cell approach allows them to monitor with great precision the temporal fluctuations in the levels of the signaling molecule, CheYp.

To some extent, this paper follows mainly the same plan as Korobkova et al. Nature 2004 and Park et al. Nature 2010. Although this paper could sound like an incremental contribution, in fact it is not. The main reason being that these initial Nature papers quantified the signaling noise in a very indirect way and have ignored the central role of the receptors; Colin et al. identified the cooperativity between receptors as a key source of the signaling noise. While I feel this paper should be published in *eLife*, I have, however, few concerns that need to be addressed to make sure that the approach is watertight and the conclusions are accurate.

1) The noise of receptorless strain still shows non-negligible noise at long time scale (Figure 1). Is the fact that the power spectrum is not flat at long timescale comes from photobleaching of the FRET pair? What is the photobleaching of the signal over the duration of the experiment? In other words what is the t_1/2_?

In any case it seems that it would be possible to normalize the power spectra of R(t) with the power spectrum from the receptorless cells in order to get rid of the unwanted noise due to photobleaching at long time scale. However, this procedure would change the current shape of the spectra at long time scale. How does photobleaching shape the power spectra of R(t) at long time scale?

2) Varying CheR level should change the relative importance of noise from the receptors versus methylation/adaptation. What is the expression level of Tar relative to [CheR] in this experiment? I am asking this because the adaptive pathway (methylation) becomes noisy only when CheR works at saturation in the ultrasensitive regime, as demonstrated in Korobkova et al. For example, if [CheR] is slightly larger than wildtype level the noise vanishes. It is therefore not enough to adjust the activity at half level to guarantee that CheR is within the saturation regime since several ratios of MeAsp/Tar/CheR can yield the same level activity. It is key in Figure 3 to know what are levels of Tar with respect to CheR levels if CheR works at saturation in order to draw a robust conclusion. Without this information, I don't find the main conclusion of the paper sufficiently supported: "These results demonstrate that long-term fluctuations in activity observed either with or without the receptor methylation system require cooperative interactions between receptors".

3) Overall I find the part of the paper on the FDT weaker because it does not bring pieces of information that we did not know before. Maybe part of it – if not all – should be move to the Appendix?

4) CheZ/CheYp cycle is not taken into account in the fluctuation-dissipation formula. The authors should either justify why or include it.

5) In order to apply the FDT, the authors should apply a stimulus small enough so that the response of the system is linear. But how do the authors make sure that they are in the linear regime? For example in Park et al. a small stepwise stimulus of 10nM of L-Aspartate (which corresponds to 1 microM of MeAsp), is close to the limit of the sensitivity of the system. Here the authors use a stimulus (>10 microM of MeAsp) that is an order of magnitude larger than that is expected to yield a linear regime. Similarly, in Martin et al. PNAS 2001 (ref 64), a great care is taken to be in the linear regime as well "We chose displacement amplitudes small enough to maintain the bundle in a regime of linear responsiveness". Again how did the authors demonstrate that 10 (or 30) microM of MeAsp stimulus would yield a linear response?

*Reviewer #3:*

The manuscript by Colin and coworkers use FRET to evaluate the dynamics of the chemotaxis system in single cells. There are several things to like about this manuscript. The authors consider one of the most interesting and well worked-out model systems in biology, and use more technically advanced imaging methods to make new discoveries. That in itself could make it of potentially broad interest. However, though I am quite familiar with stochastic analyses of bacteria, I did not find this manuscript easy to evaluate. In terms of novelty and importance I would defer to reviewers with more expertise in the specific topic. At this stage my concerns are mostly about clarification and presentation, and I do not feel comfortable recommending that the paper be accepted or rejected before these issues are explained.

First, the paper is just not written for a general audience. It jumps into the details of chemotaxis without much explanation or motivation. To make it appropriate for *eLife* would require much more explanations of the basic system, including cartoons, more descriptions of the mutants involved, stronger motivations of the questions, and a more clear emphasis of the importance of the findings. What were the surprises or big wins that scientists not working directly with chemotaxis should care about?

Second, I suspect the methods are solid, but that is hard to judge as a reader. The authors emphasize that there is a technical advancement because they no longer need to average out the FRET data over several cells, and show how the average of their single cells reproduces previous population results. That on its own in no way means that the assay is reliable in terms of the noise in single cells. I would like to see at least one paragraph describing the controls made to ensure that the noise is due to the biology, and not to imaging (heterogeneity in the evenness of excitation, camera noise etc.), to cell handling (that conditions are uniform in space and time etc.) or to reporter artefacts (that the FRET pair does not affect the circuit's behavior). Only once these controls are completed would it make sense to consider the biology.

Third, the FDT analysis is not clearly presented for a broad audience. The expected limitations of FDT analyses (thermodynamic equilibrium, or other special considerations that may or may not be satisfied in any particular system) are not really discussed, and the results will not be easy to understand for a broader audience (maybe not even for a specialized audience), e.g. the discussion of effective temperature. In my opinion the authors need to set up the question more clearly, explain the results in the language of biology or chemical kinetics rather than thermodynamics, and ensure that there is a clear narrative arc. The analysis currently comes across as a technical report for people already familiar with both the system and the analysis. I also found it odd that a recent paper from the Cluzel group was not discussed, since I believe that was the first paper showing that the fluctuation-dissipation approximation holds for some aspects of the chemotaxis system, despite an absence of thermodynamic equilibrium.

[Editors' note: further revisions were requested prior to acceptance, as described below.]

Thank you for resubmitting your work entitled "Multiple sources of slow activity fluctuations in bacterial chemosensory network" for further consideration at *eLife*. Your revised article has been favorably evaluated by Naama Barkai as the Senior and Reviewing Editor, and two reviewers.

The manuscript has been improved but there are some remaining issues that need to be addressed before acceptance, as outlined below:

Please correct the FDT analysis, or move it to the Appendix, as suggested by the reviewer. In addition, please rewrite the Abstract accordingly.

*Reviewer #1:*

In this resubmission, Colin et al. have significantly improved the readability of the paper and the theory development in the new appendix, and they added further valuable experimental controls. Although entangled in the data, I particularly like that the origins of the slow activity fluctuations could be separated into receptor dynamics and amplification by extra simulations. Overall, this is a significant contribution to chemotaxis signal transduction and quantitative cell biology as signaling noise has not been treated much so far, and I recommend publication.

*Reviewer #2:*

The main experimental results of this work are interesting and novel, and they should be published in *eLife*. The authors have clarified some of the issues of presentations that I initially had. However, the section and discussion about the application of the FDT weakens the paper because it does not bring new information about the chemotaxis system itself, and most importantly because the data do not clearly support the FDT analysis. My recommendation is to either move the section about the FDT to the Appendix and then tune down the associated claim, or maybe better to remove it entirely. The rest of the analysis with the simulations associated with Figure 5 has the merit to illustrate the different sources of noise and helps with the clarity of the paper.

1) With inset of Figure 4, the authors claim that the ratio T/T_eff_ is ~ 1, which would imply that the system is at the thermal equilibrium. I quote: "The deviation of T/T_eff_ from unity was within the range of estimations with measurement methods of similar precision": I think what this convoluted phrase means is that the deviation from 1 of the ratio T/T_eff_ is not statistically significant. But then the error bars displayed in the inset do not include the unity, which should imply that the ratio is statistically different from unity. In fact, T/T_eff_ is clearly not constant: it is about 0.5 for frequencies<10^-2 and above 1 (>1.5) at higher frequencies. Next, in inset Figure 4 the authors used a much larger scale to plot T/T_eff_ and now claim that the deviation (<=-0.5) from 0 is statistically significant and is the signature of a negative feedback loop. To conclude, the authors do not apply the same statistical standard to these two different insets, which makes the FDT analysis questionable.

By contrast, in the cited Martin, Hudspeth and Julicher, 2001 (Figure 3 in Martin et al., PNAS 2001) the ratio T/T_eff_ was randomly distributed above and below the straight line T/T_eff_=1 over an order of magnitude in frequency, and the error bars were all crossing the straight line T/T_eff_=1. In Martin et al., there is no doubt about the analysis and the interpretation of the data unlike this current work.

2) The FDT analysis may be more convincing if the authors could fit with a quantitative model the ratio T/T_eff_ like it was done for example in the Martin et al. It should be doable because the authors seem to have a full model that includes the clustering of the receptors with the adaptation pathway.

3) More generally, even if the FDT analysis were more solid, as it currently stands it does not yield new and interesting insights about the biology of the chemotaxis system. We already know that the methylation/CheA system is an energetically active process and that the cooperativity of the receptors is not. Consequently, this analysis (if proven correct) is more like a control analysis and should not hinder the main experimental results of this paper.

4) Along the same lines, the Abstract should be rewritten because it does not provide the necessary background to understand what is actually novel in this paper. As it currently stands, the abstract is a little deceptive because it suggests that the authors' study is the first one to use the FDT to analyze the chemotaxis and signaling pathways in general. For example, the concluding phrase is particularly puzzling: "We propose that such fluctuation analysis could be generally applicable to cellular networks" Of course this approach is far from being new (see Bialek et al.; PNAS; Emonet et al. PNAS; Park et al. etc…).

---

## [Author Response]

Essential revisions:As you can see from the individual reviews, all three reviewers found the subject interesting and that the work has the potential to be of broad interest. However, some significant revisions are still required.Please answer in detail all of the requests made by the reviewers. In particular, the following points were emphasized during the discussion:1) Presentation should be improved in terms of motivation, schematics, and readability.

The manuscript has been largely revised to clarify motivation, data presentation, theory development and discussion of the results and their significance in the context of published work. We notably extended and reorganized Introduction and Discussion to improve readability. We further emphasized the significance of our study in the broader context of understanding signaling noise in biochemical networks and navigation strategies of bacteria. We introduced additional explanatory figures, Figure 1—figure supplement 1 illustrating the chemotaxis pathway and the experimental setup, and Figure 5 with the graphic summary of the main results of the paper. We also modified the Results section, transferring partly redundant experimental information to supplementary figures and most of the theory development to the new Appendix, as suggested below and recommended by *eLife* guidelines.

2) Additional controls are required more make sure that noise in single-cell FRET is really due to processes inside cells, e.g. by imaging background without cells (receptor-less strain not enough).

We performed additional experiments, including the suggested negative control as well as measurements of response functions in both cheRB- and CheRB+ cells using small stimuli. Additional setup control using fluorescent beads is presented and discussed in Results and Materials and methods, as well as in new Figure 1—figure supplement 3. The verification of small stimulus assumption, as well as the new measurement of the response to small stimuli in CheRB^+^ cells, are now presented in Figure 4, as well as Figure 4—figure supplement 1 and Figure 4—figure supplement 5. This last measurement enabled to substantially improve the FDT analysis (see #3). These points are further discussed in our fourth response to reviewer 1, reviewer 2 comment 1 and reviewer 3 comment 2.

3) More rigor is needed in the theory development (best in Appendix), putting in context of previous FDT work in bacterial chemotaxis, and stressing novelty.

We improved presentation of the theory, largely extending the Appendix, and discussed more extensively the results of our analysis and their novelty and significance in the context of previously published work. To this end, Results and Discussion were significantly modified. Theory development was moved to Appendix, and reformulated to better explain our approach of comparing fluctuations and dissipation terms of the FDT to evaluate whether the system is at equilibrium or, if not, how far it deviates from equilibrium. Although the results remained unchanged for the *cheRB^-^* case, in CheRB^+^ cells, these measurements enabled to refine the FDT analysis. We also now systematically used complementary modeling approaches, coupled to a simulated toy model, to reevaluate the contribution of the methylation system and different aspects of receptor clustering to the activity fluctuations (see reviewer 1 responses 1, 4 5, 7). We explain where our work, although mostly compatible with previous results obtained using conceptually similar approaches (e.g. refs. Emonet et al. 2008, Park et al., 2010), advances the analysis of fluctuations in the chemotaxis pathway (see reviewer 1 responses 2 and 7). More generally, we reformulated discussion on how deviation from FDT can help to learn about the mechanisms behind fluctuations in relation to previous work (e.g. ref. Martin et al. 2001).

4) The main novelty is noise due to cooperative receptors, which is discussed only very briefly in paper. This claim would potentially require more elaboration (e.g. to separate receptor rearrangement from signaling noise). For the latter, receptor could be overexpressed or crosslinked to increase clustering. Theory is vague at best including Ising vs. MWC, 'adjustment' and 'rescaling' of data, and non-constant effective Temperature, and limits of applicability of FDT also need to be clarified.

We expanded the discussion of the role of receptor clustering / cooperativity in the observed activity fluctuations. Experimentally decoupling these different aspects of receptor clustering (signal amplification vs. slow receptor rearrangements) is currently not feasible, mainly because we do not fully understand the mechanism behind slow receptor rearrangements. Varying levels of receptor expression had little effect, see Figure 4—figure supplement 3, presumably because it does not sufficiently change clustering in the expression range where the response still could be measured, whereas chemical in-vivo crosslinking will most likely abolish receptor activity and have major impact on cell physiology. But, significantly modifying Results and Discussion, we have now used theoretical analysis to better understand the importance of different aspects of receptor clustering as well as the relative importance of receptor cooperativity and methylation. We have also streamlined the theory, clarified terminology and improved discussion of the FDT application, including the definition and the meaning of effective temperature. We further emphasize more clearly that deviation of the system’s behavior from the FDT is itself informative, as it enables us to determine the range of frequencies in which the active system operates.

Reviewer #1:[…] Major experimental issues:1) Subsection “Role of receptor cooperativity in signaling noise”: The noise from receptor methylation has been investigated before [Emonet and Cluzel, 2008; Park et al., 2010], so that the main experimental novelty is the signaling noise from receptor cooperativity. What is the nature of this noise – fluctuations in the signaling state or rearrangement of receptor clusters? Is there any way to separate these? This novelty section is only 10 lines long.

We have now improved the discussion of different aspects of receptor clustering that may contribute to the observed signaling noise, both in the Results and in the Discussion sections. While decoupling different contributions of clustering to signaling noise is not feasible experimentally (see #4 above), we now expanded our mathematical analysis to clarify these contributions. For clustering-dependent fluctuations in cheRB-minus cells mentioned by the reviewer, we believe that the noise arises from thermally activated slow activity dynamics of the receptor cluster, possibly coming from rearrangements within the cluster. We also believe that the signaling state and receptor arrangement are inherently connected, so that fluctuations in signaling state result from fluctuations in receptor arrangement, but at the same time activity changes drive rearrangement. The exact mechanism of this slow activity dynamics, which was observed relatively recently, is not yet fully understood. We now more clearly summarize our understanding of the nature of fluctuations in cheRB-minus as well as in CheRB+ cells in Discussion.

Subsection “Fluctuation dissipation relation for receptor clusters” on FDT: There has been significant amount of work on FDT-like properties including [Emonet and Cluzel, 2008] and [Park et al., 2010]. In the latter, they talk about the fluctuation-response theorem. What is the difference, and how does the current work fit into the context of the previous works?

We now more explicitly place our work in the context of these previous studies in the Discussion. The fluctuation-response terminology has several different applications in the literature, which we agree do not contribute to clarity: (1) in systems at equilibrium, a fluctuation-response relation is basically the zero-frequency limit of a fluctuation dissipation relation; (2) in certain classes of out-of-equilibrium systems in steady state, an extension of the FDT can be formulated, which is then called a fluctuation-response theorem; (3) following a reasoning conceptually related to the fluctuation-response theorem, refs [Emonet and Cluzel, 2008] and [Park et al., 2010] established a relation between the time scales of adaptation (called response time in these publications) and that of the fluctuations, which was termed fluctuation-response relation. As now pointed out in the Discussion, we also observe this relation, but although our FDT analysis is conceptually related to approaches (2) and (3), it is different both in its aims and technicalities, since it exploits the FDT and the system’s deviations from it to identify the noise sources in the system. The fact that FDT breaks down in CheRB^+^ case is compatible with the relation between fluctuation and adaptation time scales, since they give different, though related, information: the noise source encompasses an out-of-equilibrium process and the fluctuations originate in the chemotaxis pathway, respectively.

Figure 4 legend: The caption says that "adaptation deficient" cells were used to obtain response function in Figure 4. But the response function shown in this panel overshoots and adapts. How can these be the adaptation-deficient cells?

These cheRB- cells are indeed deficient in the methylation-based adaptation system and do not show adaptation comparable to the one observed in CheRB+ cells, so the statement “adaptation deficient” is in principle correct. However, the reviewer is right to point out that the response function in these cheRB- cells shows an interesting behavior, weakly overshooting and then “adapting”. Although weak, this behavior is reproducible (it can also be seen in Figure 2), and we believe that it results from activity-dependent slow rearrangements within the chemoreceptor clusters. We now mention this point explicitly in the text. To avoid confusion, we also now refer to these cells in the text and figure legends as “Δ*cheR*Δ*cheB”* or “lacking adaptation enzymes” rather than “adaptation deficient”.

Figure 4: These panels show power spectra and response functions for both the non-adapting (B) and adapting (C) strains, but panel A only shows response function for one strain. Where is the other response function? Panel C shows three curves but legend only mentions two-line styles. What does "rescaled" in B and "adjustment" in C really mean (in plain English)? To obtain response function, a 30 μM stimulus was apparently used, but is result really independent of stimulus strength (as long as small), or does each case need to be adjusted or rescaled separately? The theory in Materials and methods seems to be developed used an oscillating stimulus. Does it matter what the stimulus looks like?

We apologize for unclear definitions and thank the reviewer for suggesting the improvements. We have now confirmed that the response function of the cheRB- strain is essentially the same for a weaker stimulus, confirming that it is indeed independent of the stimulus strength for sub-saturating stimuli (new Figure 4—figure supplement 1). We have now also directly measured the response function of the CheRB+ strain using a weak stimulus, rather than inferring it from the response function of the cheRB- strain with the help of the adaptation model as done in the previous version of the manuscript (new Figure 4). This improved the quality of our FDT analysis for CheRB+ cells, notably by independently determining the time scale of adaptation and the fluctuation dissipation ratio in presence of adaptation (new Figure 4), and enabled us to more reliably estimate the relative contribution of different noise factors to pathway activity fluctuations in these cells (new Figure 4).

We have clarified the terminology in the figure legend. We removed confusing mentions of “rescaling” (which meant multiplication by a frequency-independent factor) by introducing the dissipation GR, which naturally appears in the FDT, and of “adjustment”, meaning a fit, which do not appear anymore in our revised treatment of the FDT. The response function in CheRB+ cells is now fitted instead (in Figure 4—figure supplement 5) to estimate the adaptation time.

We have also merged the two data sets in Figure 4 (these were data for cells expressing Tar in two different initial modification states, which give essentially identical results in the presence of the adaptation enzymes, and which we decided to consider together to improve statistics). These two data sets were also presented in the former Figure 1. One of them was moved to Figure 1—figure supplement 4 to improve readability of the manuscript, while the other remains in new Figure 1.

In the theory section, the dynamic susceptibility \chi_A(\omega) was expressed in terms of the response of *ΔcheRΔcheB* strain to a *step-like* stimulus, which was measured in Figure 4. More precisely we consider the Fourier transform of this function. We reformulated the theory development (now in the Appendix) to remove the confusing reference to an oscillating stimulus – which was meant to be the Fourier transform of the stimulus.

Figure 4: For the FDT to make any sense, T_eff_ should be constant (independent of omega), which seems to be the case when plotted as in panel D. But upon closer inspection, the blue curve changes from 2 to almost zero, so this isn't constant after all.

We assume that the reviewer refers to the cheRB- case, where the system behaves as an equilibrium one, with T_eff_ being close to T (blue curve in Figure 4). In principle, we fully agree that for a perfectly measured equilibrium system T_eff_ should simply equal T, and T_eff_/T = 1 over the entire frequency range. However, experimental measurements of T_eff_ are typically imprecise, and the observed deviation from unity is within the range of data reported for other systems with similar precisions (e.g., refs Abou and Gallet PRL 2004, Wang, Song and Makse Nature Physics (2006), Martin et al. PNAS (2001)). We now explicitly discuss this issue in the text. To facilitate the comparison with ideal behavior, we now added a dashed line representing T_eff_ = T in the inset of new Figure 4, where these data are now presented.

In contrast, for an out-of-equilibrium system, T_eff_ may depend on omega, since the FDT breaks down only in a particular frequency range. This could be seen, for example, in the previous analysis of the inner ear hair bundle (ref Martin et al., PNA*S* (2001)). Notably, upon improving our analysis by using the measured rather than inferred response function for CheRB+ cells (see our fourth response to reviewer 1), the behavior of T_eff_ for CheRB+ cells now actually resembles that reported by both Martin et al., PNAS (2001) and also predicted by Sartori and Tu, PRL (2015), turning negative when the active process (CheRB-dependent methylation in our case), which is a delayed – slow – negative feedback in all cases, overcomes the passive behavior of the system at low frequency, and remaining positive at higher frequencies. The frequency at which this occurs reflects this interplay between passive behavior and negative feedback. We now elaborate on the question of frequency dependence of T_eff_ both in the main text (Results and Discussion) and in the Appendix, making comparisons to this and other previous works.

Major theoretical issues:Introduction, fifth paragraph and subsection “Modeling activity fluctuations in the framework of fluctuation dissipation relation”, first two paragraphs (Materials and methods): both the Ising and the MWC model are used. Why both? Why not just use the simpler MWC model?

As MWC is a mean field model, it doesn’t contain fluctuations, although it can be formulated to include a certain type of fluctuations as done now in the Appendix, in particular for simple simulations of the receptor clusters. In this case, only the signaling teams fluctuate as a whole, which neglect fluctuations of individual receptors and kinases, being therefore not very satisfactory. On the other hand, the Ising model naturally incorporates activity fluctuations in its formulation, and it was thus favored in our theory development. These differences are now plainly explained in the theory development. In general, we have now stronger focused the theory on the Ising model.

Subsection “Fluctuation dissipation relation for receptor clusters” on FDT: A major issue is that the theory is only described very briefly in the Materials and methods section, and that it is almost impossible to understand. Why is it not derived in detail, e.g. in the Appendix (which is currently only one page long). When looking through the literature, found paper Clausnitzer and Enders, BMC SB (2011), which also seems to derive response functions and signaling noises. How does the current work relate to their theory and predictions?

The theory development was moved to Appendix and described more extensively. We apologize for not mentioning the study pointed out by the referee, it is now properly acknowledged and discussed in relation to our work.

Another theory paper, [Sartori and Tu, 2015], says that the FDT breaks down or is violated in the chemotaxis pathway. Why is the FDT then used to study the chemotaxis pathway? Why do power spectra and response functions even match? Maybe the rescaling/adjustment really means that they don't match.

Again, we apologize for not being sufficiently clear in explaining the motivation for using the FDT and the interpretation of the results. Our prime reason to apply FDT was to understand the nature of fluctuations in cheRB- and CheRB+ cells, and thus the deviation from the FDT is actually informative. For cheRB- cells, our analysis showed that the FDT is satisfied, meaning that the system in this case behaves as in equilibrium. This was an important and absolutely non-trivial result. Even for CheRB+ cells, where the violation of the FDT was indeed theoretically predicted in Sartori and Tu, PRL (2015), application of the FDT-based analysis directly demonstrated this violation and showed that the system is out of equilibrium in a particular frequency range. The rational and the conclusions of the FDT application are now better spelled out in the Results and Discussion.

Reviewer #2:This paper aims at quantifying beside the methylation/adaptive system whether receptors can contribute to the signaling noise of the chemotaxis network in E. coli. The authors use a robust technique based on the tandem FRET CheY-YFP and CheZ-CFP that has been well-established about more than 10 years ago. The technical improvement from older studies by the same group is that they now use a CCD camera to monitor fluctuations in the Fret signal at the single cell level. This single-cell approach allows them to monitor with great precision the temporal fluctuations in the levels of the signaling molecule, CheYp.To some extent, this paper follows mainly the same plan as Korobkova et al. Nature 2004 and Park et al. Nature 2010. Although this paper could sound like an incremental contribution, in fact it is not. The main reason being that these initial Nature papers quantified the signaling noise in a very indirect way and have ignored the central role of the receptors; Colin et al. identified the cooperativity between receptors as a key source of the signaling noise. While I feel this paper should be published in eLife, I have, however, few concerns that need to be addressed to make sure that the approach is watertight and the conclusions are accurate.1) The noise of receptorless strain still shows non-negligible noise at long time scale (Figure 1). Is the fact that the power spectrum is not flat at long timescale comes from photobleaching of the FRET pair? What is the photobleaching of the signal over the duration of the experiment? In other words what is the t_1/2_?In any case it seems that it would be possible to normalize the power spectra of R(t) with the power spectrum from the receptorless cells in order to get rid of the unwanted noise due to photobleaching at long time scale. However, this procedure would change the current shape of the spectra at long time scale. How does photobleaching shape the power spectra of R(t) at long time scale?

Although the curve for the receptorless strain is mostly flat, as expected for a white shot noise, it does show a weak increase at low frequency as rightly pointed out by the Reviewer. We now additionally present YFP/CFP ratios measured with beads having a large fluorescence spectrum, for which bleaching is completely negligible, and which emit in both the YFP and CFP filter windows. This confirmed that instrumental noise results in pure white noise for the YFP/CFP ratio. The weak low frequency dependence might indeed come from photobleaching, although other phenomena like slow fluctuations in the cell state could also play an additional role, as now mentioned in the text.

Irrespective of the precise origin of this deviation from the flat baseline, it makes only minor contribution to the PSD of strains with intermediate activity (one has to consider that all PSDs are plotted in log scale). To illustrate it, we now show the PSD curves for which the noise contribution was subtracted in new Figure 1—figure supplement 4 and Figure 2—figure supplement 1. Note that constant noise background is accounted for in the FDT formulation, as explicitly seen in Equation (2). These points are now discussed in Results and Materials and methods.

2) Varying CheR level should change the relative importance of noise from the receptors versus methylation/adaptation. What is the expression level of Tar relative to [CheR] in this experiment? I am asking this because the adaptive pathway (methylation) becomes noisy only when CheR works at saturation in the ultrasensitive regime, as demonstrated in Korobkova et al. For example, if [CheR] is slightly larger than wildtype level the noise vanishes. It is therefore not enough to adjust the activity at half level to guarantee that CheR is within the saturation regime since several ratios of MeAsp/Tar/CheR can yield the same level activity. It is key in Figure 3 to know what are levels of Tar with respect to CheR levels if CheR works at saturation in order to draw a robust conclusion. Without this information, I don't find the main conclusion of the paper sufficiently supported: "These results demonstrate that long-term fluctuations in activity observed either with or without the receptor methylation system require cooperative interactions between receptors".

We should have indeed mentioned this issue more explicitly. In all our experiments (except those in Figure 4—figure supplement 2 were Tar induction level was systematically varied), Tar was expressed at approximately 10^4^ dimers per cell, which is similar to the total level of receptor expression in wild type. The level of Tar expression was previously mentioned but only when discussing theory development; we now explicitly state it at the beginning of the Results section. We believe that CheR and CheB are expressed at their native levels (few hundred copies per cell), including in CheW-Χ2 strains. Indeed in this strain, even CheW itself is expressed at the wild-type level (Pinas et al. PNAS (2016)), and we thus do not expect that this point mutation on a structural protein with no known gene regulatory function could affect the expression of genes expressed from a different operon. Thus the receptor is indeed in large excess relative to CheR, as it is in the wild-type cells, and therefore CheR is expected to work at saturation. We now clarify this point in the Results and Discussion.

3) Overall I find the part of the paper on the FDT weaker because it does not bring pieces of information that we did not know before. Maybe part of it – if not all – should be move to the Appendix?

Part of the FDT formulation was moved to Appendix, as suggested. We have further extended the analysis and spelled out our motivation more clearly.

4) CheZ/CheYp cycle is not taken into account in the fluctuation-dissipation formula. The authors should either justify why or include it.

Since the dynamics of the CheZ/CheYP cycle takes place on a time scale shorter than the second, below the time resolution of our measurements, it is not expected to contribute significantly to the observed noise. Consistent with that, no fluctuations above the shot noise background is experimentally observed at high frequencies. Moreover CheY and CheZ are abundant and not cooperative, we therefore expect they contribute very little as a noise source. We now discuss it during the theory development in Appendix and in Discussion.

5) In order to apply the FDT, the authors should apply a stimulus small enough so that the response of the system is linear. But how do the authors make sure that they are in the linear regime? For example in Park et al. a small stepwise stimulus of 10nM of L-Aspartate (which corresponds to 1 microM of MeAsp), is close to the limit of the sensitivity of the system. Here the authors use a stimulus (>10 microM of MeAsp) that is an order of magnitude larger than that is expected to yield a linear regime. Similarly, in Martin et al. PNAS 2001 (ref 64), a great care is taken to be in the linear regime as well "We chose displacement amplitudes small enough to maintain the bundle in a regime of linear responsiveness". Again how did the authors demonstrate that 10 (or 30) microM of MeAsp stimulus would yield a linear response?

See our fourth response to reviewer 1.

Reviewer #3:The manuscript by Colin and coworkers use FRET to evaluate the dynamics of the chemotaxis system in single cells. There are several things to like about this manuscript. The authors consider one of the most interesting and well worked-out model systems in biology, and use more technically advanced imaging methods to make new discoveries. That in itself could make it of potentially broad interest. However, though I am quite familiar with stochastic analyses of bacteria, I did not find this manuscript easy to evaluate. In terms of novelty and importance I would defer to reviewers with more expertise in the specific topic. At this stage my concerns are mostly about clarification and presentation, and I do not feel comfortable recommending that the paper be accepted or rejected before these issues are explained.First, the paper is just not written for a general audience. It jumps into the details of chemotaxis without much explanation or motivation. To make it appropriate for eLife would require much more explanations of the basic system, including cartoons, more descriptions of the mutants involved, stronger motivations of the questions, and a more clear emphasis of the importance of the findings. What were the surprises or big wins that scientists not working directly with chemotaxis should care about?

We apologize for not sufficiently elaborating on general motivation behind our study and on the significance of our findings, and we agree on importance of making our manuscript better accessible to scientists that are not well familiar with bacterial chemotaxis. The revised version of the manuscript addresses these issues, as well as giving more thorough clarification of the mutants and theoretical and experimental methodology.

Second, I suspect the methods are solid, but that is hard to judge as a reader. The authors emphasize that there is a technical advancement because they no longer need to average out the FRET data over several cells, and show how the average of their single cells reproduces previous population results. That on its own in no way means that the assay is reliable in terms of the noise in single cells. I would like to see at least one paragraph describing the controls made to ensure that the noise is due to the biology, and not to imaging (heterogeneity in the evenness of excitation, camera noise etc.), to cell handling (that conditions are uniform in space and time etc.) or to reporter artefacts (that the FRET pair does not affect the circuit's behavior). Only once these controls are completed would it make sense to consider the biology.

As mentioned above in our first response to reviewer 3, we revised the manuscript to give more details on the methodology, including details of FRET measurements and their interpretation. We have also performed additional control experiments, confirming that observed low-frequency fluctuations are indeed due to biology, and elaborated on the distinction between these biological fluctuations seen in certain strains and measurement noise that is present in all measured strains and under all conditions (see also our first response to reviewer 2).

Third, the FDT analysis is not clearly presented for a broad audience. The expected limitations of FDT analyses (thermodynamic equilibrium, or other special considerations that may or may not be satisfied in any particular system) are not really discussed, and the results will not be easy to understand for a broader audience (maybe not even for a specialized audience), e.g. the discussion of effective temperature. In my opinion the authors need to set up the question more clearly, explain the results in the language of biology or chemical kinetics rather than thermodynamics, and ensure that there is a clear narrative arc. The analysis currently comes across as a technical report for people already familiar with both the system and the analysis. I also found it odd that a recent paper from the Cluzel group was not discussed, since I believe that was the first paper showing that the fluctuation-dissipation approximation holds for some aspects of the chemotaxis system, despite an absence of thermodynamic equilibrium.

We now expanded the Results and Discussion to better discuss the FDT and significance of its application to the chemotaxis system, also in the context of published literature (also see our second and seventh and last responses to reviewer 1).

[Editors' note: further revisions were requested prior to acceptance, as described below.]

Reviewer #2:The main experimental results of this work are interesting and novel, and they should be published in eLife. The authors have clarified some of the issues of presentations that I initially had. However, the section and discussion about the application of the FDT weakens the paper because it does not bring new information about the chemotaxis system itself, and most importantly because the data do not clearly support the FDT analysis. My recommendation is to either move the section about the FDT to the Appendix and then tune down the associated claim, or maybe better to remove it entirely. The rest of the analysis with the simulations associated with Figure 5 has the merit to illustrate the different sources of noise and helps with the clarity of the paper.

We thank the Reviewer for acknowledging the novelty of our work and supporting its publication. We have now addressed remaining questions related to the FDT. Most importantly, we have shortened the FDT section and discussed the deviations from the FDT. Because the FDT is an integral component of the analysis that enabled us to decompose the two noise sources, importance of which is recognized by both reviewers, we felt that it is better to keep the essential part of the FDT analysis in the main text, to enable readers to follow the analysis.

1) With inset of Figure 4, the authors claim that the ratio T/T_eff_ is ~ 1, which would imply that the system is at the thermal equilibrium. I quote: "The deviation of T/T_eff_ from unity was within the range of estimations with measurement methods of similar precision": I think what this convoluted phrase means is that the deviation from 1 of the ratio T/T_eff_ is not statistically significant. But then the error bars displayed in the inset do not include the unity, which should imply that the ratio is statistically different from unity. In fact, T/T_eff_ is clearly not constant: it is about 0.5 for frequencies<10^-2 and above 1 (>1.5) at higher frequencies. Next, in inset Figure 4 the authors used a much larger scale to plot T/T_eff_ and now claim that the deviation (<=-0.5) from 0 is statistically significant and is the signature of a negative feedback loop. To conclude, the authors do not apply the same statistical standard to these two different insets, which makes the FDT analysis questionable.By contrast, in the cited Martin, Hudspeth and Julicher, 2001 (Figure 3 in Martin et al., PNAS 2001) the ratio T/T_eff_ was randomly distributed above and below the straight line T/T_eff_=1 over an order of magnitude in frequency, and the error bars were all crossing the straight line T/T_eff_=1. In Martin et al., there is no doubt about the analysis and the interpretation of the data unlike this current work.

We agree that in the cheRB- strain, the deviation of the FDT ratio from unity is not negligible, and it might indicate contributions of out-of-equilibrium cellular processes to the receptor array stimulation. Nevertheless T/T_eff_ ratio in this strain remains fairly close to unity, in contrast to T/T_eff_ ratio in the CheRB+ strain that is far from unity and even negative over a part of the frequency spectrum. To emphasize this difference, we now set the insets of Figure 4 to the same scale, as also suggested by the reviewer. Thus thermal noise is clearly the main contributor to activity fluctuations in the cheRB- strain, despite possible second-order contributions from other sources. We modified Results (subsection “Fluctuation-dissipation relation for receptor clusters”, fourth paragraph) and Discussion (fifth paragraph) to clarify this point.

2) The FDT analysis may be more convincing if the authors could fit with a quantitative model the ratio T/T_eff_ like it was done for example in the Martin et al. It should be doable because the authors seem to have a full model that includes the clustering of the receptors with the adaptation pathway.

The analytical model we used to describe the effect of the slow dynamics of receptor clusters is purely phenomenological, because there is currently no mechanistic explanation for slow cluster rearrangements in the cheRB- strains. The simulations thus included this slow dynamic in a very heuristic manner, meaning that agreement between simulations and experiments is only qualitative, as we already pointed out in the Results (subsection “Out-of-equilibrium dynamics in presence of adaptation system”, last paragraph) and the appendix (Appendix section 3 **“**Separating the contribution of methylation enzymes dynamics to the PSD in CheR^+^ CheB^+^ cells” and following). We now reemphasized these points in the Results (subsection “Fluctuation-dissipation relation for receptor clusters”, last paragraph) and Discussion (sixth paragraph). Furthermore such modeling would require knowledge of the dynamics of the fluctuations of the activity of the adaptation enzymes, which had not been previously described to our knowledge. We therefore preferred to analyze the activity fluctuations in terms of equilibrium and out-of-equilibrium components, as carried out in Figure 4), to separate the contribution of thermally activated receptor noise and adaptation enzyme noise, and then infer the fluctuation dynamics of the methylation enzymes.

3) More generally, even if the FDT analysis were more solid, as it currently stands it does not yield new and interesting insights about the biology of the chemotaxis system. We already know that the methylation/CheA system is an energetically active process and that the cooperativity of the receptors is not. Consequently, this analysis (if proven correct) is more like a control analysis and should not hinder the main experimental results of this paper.

Although we indeed did not expect the cooperative chemoreceptor cluster to have a build-in active system, it was not obvious and actually surprising to us that thermal fluctuations alone were enough to account for a large part of the observed fluctuations. As a matter of fact, our starting hypothesis was an involvement of some hypothetical non-equilibrium process in driving fluctuations in the cheRB- strain. This is now mentioned when justifying the application of the FDT in the Results (subsection “Fluctuation-dissipation relation for receptor clusters”, first paragraph). Thus, the finding that thermal noise could drive fluctuations of such magnitude, making a large part of the array to flip activity, is one of the main and most surprising results of this paper, and we re-emphasized it now in the Discussion (fifth paragraph).

In the CheRB+ case, although it was indeed expected that FDT breaks down, the computations of fluctuation and response functions are however a necessary preliminary to the separation of noise sources. Therefore, the FDT analysis is a relevant part of this paper, which integrates into its narrative. We however shortened this part, by moving technicalities and less relevant information to supplementary (new Figure 4—figure supplement 8, parts moved from Results to Discussion, last paragraph and Appendix), in an effort to highlight the most relevant results.

4) Along the same lines, the Abstract should be rewritten because it does not provide the necessary background to understand what is actually novel in this paper. As it currently stands, the abstract is a little deceptive because it suggests that the authors' study is the first one to use the FDT to analyze the chemotaxis and signaling pathways in general. For example, the concluding phrase is particularly puzzling: "We propose that such fluctuation analysis could be generally applicable to cellular networks" Of course this approach is far from being new (see Bialek et al.; PNAS; Emonet et al. PNAS; Park et al. etc…).

We rewrote the Abstract to emphasize the novelty of the manuscript. It was certainly not our intension to claim that our study is the first one to use fluctuation dissipation relations to infer properties of a biological system, and we therefore refer extensively to some of the most prominent previous work in the Introduction and Discussion section. However, there are important differences: Some of these previous works, from which we drew the methodology used in this work (primarily Martin et al), were investigating entirely different systems. Others (Korobkova et al., Park et al., Emonet and Cluzel) were interested in the chemotaxis pathway, but followed a different methodology and were interested in different quantities than the one we considered. We now discuss the results of previous studies and the relation between our and the previous works, also referring to Bialek et al. PNAS 2005, in greater detail in the Discussion (fourth and last paragraphs). As mentioned above, we however believe that the uncovering of noise sources and their relative importance yielded by the present FDT analysis is one of the novelties of our paper.